

# Comparative assessment of TROPOMI and OMI formaldehyde observations against MAX-DOAS network column measurements.

Isabelle De Smedt[1], Gaia Pinardi[1], Corinne Vigouroux[1], Steven Compernolle[1], Alkis Bais[2], Nuria Benavent[3], Folkert Boersma[4,5], Ka-Lok Chan[6], Sebastian Donner[7], Kai-Uwe Eichmann[8], Pascal Hedelt[6], François Hendrick[1], Hitoshi Irie[9], Vinod Kumar[7], Jean-Christopher Lambert[1], Bavo Langerock[1], Christophe Lerot[1], Cheng Liu[10], Diego Loyola[6], Ankie Piters[4], Andreas Richter[8], Claudia Rivera Cárdenas[11], Fabian Romahn[6], Robert George Ryan[12,13], Vinayak Sinha[14], Nicolas Theys[1], Jonas Vlietinck[1], Thomas Wagner[7], Ting Wang[15], Huan Yu[1], Michel Van Roozendael[1].

*Correspondence to*: Isabelle De Smedt (isabelle.desmedt@aeronomie.be)

1. Royal Belgian Institute for Space Aeronomy (BIRA-IASB), Ringlaan 3, 1180 Uccle, Belgium.
2. Laboratory of Atmospheric Physics, Aristotle University of Thessaloniki (AUTH), Thessaloniki, Greece.
3. Department of Atmospheric Chemistry and Climate, Institute of Physical Chemistry Rocasolano (CSIC), Madrid, Spain.
4. Royal Netherlands Meteorological Institute (KNMI), De Bilt, the Netherlands.
5. Meteorology and Air Quality group, Wageningen University, the Netherlands.
6. Institut für Methodik der Fernerkundung (IMF), Deutsches Zentrum für Luft und Raumfahrt (DLR), Oberpfaffenhofen, Germany.
7. Max-Planck-Institut für Chemie (MPI-C), Mainz, Germany.
8. Institute of Environmental Physics, University of Bremen (IUP-B), Bremen, Germany.
9. Center for Environmental Remote Sensing, Chiba University (Chiba U), Chiba, Japan
10. Department of Precision Machinery and Precision Instrumentation, University of Science and Technology of China, Hefei, China.
11. Centro de Ciencias de la Atmósfera, Universidad Nacional Autónoma de México (UNAM), Mexico City, Mexico
12. School of Earth Sciences, The University of Melbourne, Melbourne, Australia
13. ARC Centre of Excellence for Climate System Science, Sydney, Australia
14. Department of Earth and Environmental Sciences, Indian Institute of Science Education and Research (IISER), Mohali, India
15. Institute of Atmospheric Physics, Chinese Academy of Sciences (CAS), Beijing, China

**Abstract.** The TROPOspheric Monitoring Instrument (TROPOMI), launched in October 2017 on board the Sentinel-5 Precursor (S5P) satellite, monitors the composition of the Earth's atmosphere at an unprecedented horizontal resolution as fine as 3.5x5.5 km$^2$. This paper assess the performances of the TROPOMI formaldehyde (HCHO) operational product compared to its predecessor, the OMI HCHO QA4ECV product, at different spatial and temporal scales. The parallel development of the two algorithms favored the consistency of the products, which facilitates the production of long-term combined time series. The main difference between the two satellite products is related to the use of different cloud algorithms, leading to a positive bias of OMI compared to TROPOMI of up to 30% in Tropical regions. We show that after switching off the explicit correction for cloud effects, the two datasets come into an excellent agreement. For medium to large HCHO vertical columns (larger than 5x10$^{15}$ molec.cm$^{-2}$) the median bias between OMI and TROPOMI HCHO columns is not larger than 10% (<0.4x10$^{15}$ molec.cm$^{-2}$). For lower columns, OMI observations present a remaining positive bias of about 20% (<0.8x10$^{15}$ molec.cm$^{-2}$) compared to TROPOMI in mid-latitude regions. Here, we also use a global network of 18 MAX-DOAS instruments to validate both satellite sensors for a large range of HCHO columns. This work complements the study by Vigouroux et al. (2020) where a



global FTIR network is used to validate the TROPOMI HCHO operational product. Consistent with the FTIR
validation study, we find that for elevated HCHO columns, TROPOMI data are systematically low (-25% for HCHO
columns larger than $8 \times 10^{15}$ molec.cm$^{-2}$), while no significant bias is found for medium range column values. We
further show that OMI and TROPOMI data present equivalent biases for large HCHO levels. However, TROPOMI
significantly improves the precision of the HCHO observations at short temporal scales, and for low HCHO columns.
We show that compared to OMI, the precision of the TROPOMI HCHO columns is improved by 25% for individual
pixels, and up to a factor 3 when considering daily averages in 20km-radius circles. The validation precision obtained
with daily TROPOMI observations is comparable to the one obtained with monthly OMI observations. To illustrate
the improved performances of TROPOMI in capturing weak HCHO signals, we present clear detection of HCHO
column enhancements related to shipping emissions in the Indian Ocean. This is achieved by averaging data over a
much shorter period (3 months) than required with previous sensors, and opens new perspectives to study shipping
emissions of VOCs and related atmospheric chemical interactions.
## 1    Introduction
Satellite observations of tropospheric formaldehyde (HCHO) columns have been used for years to support air quality
and chemistry-climate related studies from the regional to the global scale. Formaldehyde is an intermediate gas in
almost all oxidation chains of non-methane volatile organic compounds (NMVOC), leading to the production of CO,
and eventually $CO_2$. NMVOCs are, together with NOx, CO and $CH_4$, among the most important precursors of
tropospheric ozone. NMVOCs also produce secondary organic aerosols and influence the concentrations of OH, the
main tropospheric oxidant. The major HCHO source in the remote atmosphere is $CH_4$ oxidation. Over the continents,
the oxidation of other NMVOCs emitted from vegetation, fires, traffic and industrial sources results in important and
localised enhancements of the HCHO levels. Because its short lifetime (of the order of a few hours), HCHO in the
boundary layer can be related to the release of a large number of short-lived volatile hydrocarbons. Furthermore,
HCHO observations provide information on the chemical oxidation processes in the atmosphere, including CO
chemical production from $CH_4$ and NMVOC, the oxidation of isoprene into HCHO, which allows quantification of
midday OH (Wells et al., Nature, 2019), and the tropospheric ozone production regimes that depend on the HCHO to
$NO_2$ ratios (Jin et al., 2020).
Satellite observations of formaldehyde columns in the troposphere have been extensively reported in the literature
from a number of nadir UV sensors, e.g.: Global Ozone Monitoring Experiment (GOME; Chance et al., 2000; Palmer
et al., 2001; De Smedt et al., 2008), SCanning Imaging Absorption spectroMeter for Atmospheric CHartographY
(SCIAMACHY; Wittrock et al., 2006; De Smedt et al., 2008; 2010), Ozone Monitoring Instrument (OMI; González
Abad et al., 2015; De Smedt et al., 2015; 2018; Kaiser et al. 2018; Levelt et al., 2018), Global Ozone Monitoring
Experiment-2 (GOME-2; De Smedt et al., 2012; 2015; Vrekoussis et al., 2010; Hewson et al., 2013; Hassinen et al.,
2016), and Ozone Mapping and Profiler Suite (OMPS; Li et al., 2015; González Abad et al., 2016). They are used in
many studies related to air quality and climate change (e.g. Stavrakou et al., 2014; 2015; 2016; 2018; Fortems-Cheiney
et al., 2012; Marais et al., 2012; Mahajan et al., 2015; Choi et al., 2015; Zhu et al., 2016; Chan Miller et al., 2017; Jin





et al., 2017; Barkley et al., 2017; Cao et al., 2018; Khan et al., 2018; Surl et al., 2018; Shen et al. 2019; Su et al.; 2019;
Zyrichidou et al., 2019; Jin et al., 2020; Souri et al., 2020; Wells et al., 2020; Franco et al., 2021; Opacka et al., 2021).
Launched on board of the European Copernicus Sentinel-5 Precursor (S5P) satellite on 13 October 2017, the
TROPOspheric Monitoring Instrument (TROPOMI, Veefkind et al., 2012) is designed for the daily monitoring of the
troposphere at the global scale. Compared to its predecessor OMI, its spatial resolution is about 16 times better with
at least the same signal to noise ratio per ground pixel. The improved TROPOMI capabilities for the observation of
HCHO have been illustrated for the detection of fire plumes and their transport (Alvarado et al., 2020; Theys et al.
2020), and the detection of rapid changes in anthropogenic emissions related to the COVID crisis in China and India
(Levelt et al., 2021; Sun et al. 2021). The TROPOMI observations extend the historical time series of midday
observations performed using OMI. Both datasets are used in combination for long-term trend studies (Li et al., 2020).
It is therefore important to evaluate their level of agreement and to report on the best practices to combine datasets
from different sensors.
The TROPOMI vertical column product requirements specify a single measurement precision of $12 \times 10^{15}$ molec.cm$^{-2}$,
$4 \times 10^{15}$ molec.cm$^{-2}$ at 20km spatial resolution, and a systematic uncertainty lower than 40%-80% (ESA, 2014). The
Copernicus user requirements, primarily defined for NMVOC measurements, are more stringent. For the
environmental air quality theme, the required maximum uncertainty is defined as 60% or $1.3 \times 10^{15}$ molec.cm$^{-2}$ (least
stringent), at the spatial resolution of 20km and with a revisit time of 2 hours. The space and time resolution are less
stringent for the climate theme (30% or $1.3 \times 10^{15}$ molec.cm$^{-2}$, 50km, 3 days) (Bovensmann et al., 2011; Langen et al.,

96    2017).

Given these rather strict product requirement and the diversity of the NMVOC species, lifetimes and sources (biogenic,
biomass burning or anthropogenic), a validation approach addressing a large variety of conditions worldwide (tropical,
temperate and boreal forests, urban and sub-urban areas) is needed, as well as continuous measurements in order to
obtain good statistics and capture the seasonal variations. Vigouroux et al. (2020) validated the operational TROPOMI
HCHO product using a global network of Fourier Transform Infrared (FTIR) instruments. The study concluded that
overall the HCHO product fulfils the requirements of the TROPOMI mission. Compared to the FTIR data, the
TROPOMI HCHO columns present a negative bias over high emission sites (-31% for HCHO columns larger than
$8 \times 10^{15}$ molec.cm$^{-2}$) and a positive bias for clean sites (+26% for HCHO columns lower than $2.5 \times 10^{15}$ molec.cm$^{-2}$).
Based on clean sites, an upper limit of $1.3 \times 10^{15}$ molec.cm$^{-2}$ was estimated for the deviation of daily observations at a
spatial resolution of 20km. It was also pointed out that this level of random uncertainty, although reaching the
Copernicus user requirements, is about twice as large as the expected theoretical noise (individual pixel precision
divided by the square root of the number of observations). However, Vigouroux et al. (2020) do not address the
consistency of TROPOMI HCHO with other satellite products and MAX-DOAS HCHO observations.
The present paper is a follow-up of De Smedt et al. (2018), where the HCHO retrieval algorithm applied to both OMI
and TROPOMI sensors was presented. Here we concentrate on a global study of three years of HCHO observations
with TROPOMI, and we analyse their consistency with OMI data. Throughout the paper, we discuss the improved
capabilities of TROPOMI for the detection of HCHO at different temporal and spatial scales, from background
conditions to high emissions. We start with a few illustrations of the TROPOMI capabilities for HCHO monitoring





from space (sect. 3). We then provide a detailed comparison with the OMI QA4ECV HCHO dataset (sect. 4). In sect.
5, a global network of MAX-DOAS instruments is used to validate the OMI and TROPOMI HCHO datasets. Finally,
in sect. 6, we illustrate the enhanced capability of TROPOMI for the detection of very small HCHO emissions with
the identification of a signal over shipping lanes in the Indian Ocean.
**2**     **HCHO Datasets**
**2.1**     **OMI instrument and QA4ECV HCHO product**
The Aura satellite was launched in July 2004, in a low-Earth polar orbit crossing the equator at 13:30 LT. On board
of Aura, the Ozone Monitoring Instrument (OMI) is a nadir viewing imaging spectrometer that measures the solar
radiation backscattered by the Earth's atmosphere and surface over the wavelength range from 270 to 500 nm (Levelt
et al., 2006). Operational Level 2 (L2) products include vertical columns of $O_3$, $SO_2$, $NO_2$, HCHO, BrO, OClO, as
well as cloud and aerosol information. OMI has a 2600 km wide swath (divided into 60 across-track positions or
rows), providing near-daily global coverage. However, due to a detector row anomaly that occurred after a few years
of operation, an increasing number of rows had to be filtered out leading to gradual degradation of the coverage. The
OMI ground pixel size varies from 13x24 km² at nadir to 28x150 km² at the edges of the swath.
The OMI QA4ECV HCHO product was developed by a European consortium (BIRA, IUP, MPIC, KNMI, WUR) (De
Smedt et al., 2017, http://doi.org/10.18758/71021031) in the framework of the EU-FP7 QA4ECV project. A detailed
step-by-step study was performed for HCHO and $NO_2$ retrievals as part of a community effort to homogenize GOME,
SCIAMACHY, GOME-2 and OMI, leading to state-of-the art European products (www.qa4ecv.eu). For this study,
we use the version 1.2 of the OMI HCHO dataset that is now spanning 15 years (2005-2020; Boersma et al., 2018;
Lorente et al., 2017; Nightingale et al., 2018; Zara et al., 2018). Note that within QA4ECV, a homogenized dataset of
$NO_2$ and HCHO MAX-DOAS reference measurements (QA4ECV_MAXDOAS) was also developed for satellite
validation (see sect. 2.4 and sect. 5).
**2.2**     **TROPOMI instrument and the HCHO operational product**
On board of the S5P platform, which - like Aura - flies in a low-Earth afternoon polar orbit with a local overpass time
of 13:30, the TROPOMI instrument is based on an imaging spectrometer measuring in the ultraviolet (UV), visible
(VIS), near-infrared (NIR), and shortwave infrared (SWIR) spectral regions (Veefkind et al., 2012). Operational L2
products include vertical columns of $O_3$, $SO_2$, $NO_2$, HCHO, CO and $CH_4$, as well as cloud and aerosol information.
TROPOMI has a 2600 km wide swath (divided into 450 across-track positions or rows), providing near-daily global
coverage. The spatial resolution at nadir, originally of 3.5x7 $km^2$ (across-track x along-track) has been refined to
3.5x5.5 $km^2$ on 6 August 2019, by a change in the along-track integration time. The size of the pixels remains more
or less constant towards the edges of the swath (the largest pixels are ~14 km wide) (L1b ATBD, L1b readme file).
The retrieval algorithm of the TROPOMI HCHO L2 product is directly inherited from the QA4ECV OMI algorithm
with the aim to create a consistent time series of early afternoon observations. For this study, we use a modified version
of the TROPOMI level-2 HCHO operational data product, which starts in April 2018 (phase E2, RPRO+OFFL,



product versions 1.1.[5-8]+2.1.3, doi: 10.5270/S5P-tjlxfd2). Product versions are described in the Product Readme
File.

## 2.3    HCHO Retrieval algorithm for OMI and TROPOMI

The HCHO retrieval algorithm was fully described in De Smedt et al. (2018), and the successive adaptations of the
algorithm are reported in the S5P product ATBD. Here we only provide a short description of the algorithm, which is
based on a 3-steps DOAS method. First, the fit of the slant columns ($N_s$) is performed in the UV part of the spectra,
in the fitting interval 328.5-359 nm. The HCHO cross-section is from Meller and Moortgat (2000). All cross-sections
have been pre-convolved for every row separately with an instrumental slit function adjusted after TROPOMI launch.
For the OMI product, the slit function of each row is adjusted daily and the cross-sections are reconvolved accordingly.
The DOAS reference spectrum is updated daily with an average of Earth radiances measured in the Equatorial Pacific
region from the previous day. The fit therefore results in a differential slant column, corresponding to the HCHO
excess over sources compared to the remote background. In a second step, the conversion from slant to tropospheric
vertical columns ($N_v$) is performed using a look up table of vertically resolved air mass factors ($M$) calculated at 340
nm with the radiative transfer model VLIDORT v2.6 (Spurr, 2008). Entries for each ground pixel are the observation
geometry, the surface elevation and reflectivity, as well as clouds treated as reflecting surfaces, and a priori
tropospheric HCHO profiles. The surface albedo is taken from the monthly OMI albedo climatology at the spatial
resolution of 1° x1° (minimum LER, Kleipool et al., 2008). A priori vertical profiles are provided by the TM5-MP
daily analysis, at the spatial resolution of 1°x1° (Williams et al., 2017). A cloud correction based on the independent
pixel approximation (Boersma et al., 2004) is applied for cloud fractions (CF) larger than 0.1. Finally, to correct for
any remaining global offset and possible stripes arising between the rows, a background correction is performed based
on the HCHO slant columns in the Pacific Ocean ($N_{s,0}$). For the TROPOMI operational product, $N_{s,0}$ is based on the
four previous days. For this study, and for the OMI product, we perform the correction on the current day in order to
further reduce the stripes. To compensate for a background HCHO level in the Equatorial Pacific (due to the methane
oxidation), a vertical column of HCHO ($N_{v,0}^{CTM}$) is taken from the TM5 model in the reference region. The resulting
tropospheric HCHO vertical column can be written as follows:

$$N_v = \frac{N_s - N_{s,0}}{M} + \frac{M_0}{M} N_{v,0}^{CTM}, \qquad\qquad (2\text{-}1)$$

with $M_0$ the air mass factor in the reference sector. Intermediate quantities and auxiliary data are all stored in the L2
files (see the product user manual for TROPOMI and OMI). Several diagnostic variables are provided together with
the measurements. The column averaging kernels and the a priori profiles are given for each observation. The
tropospheric column uncertainty is resolved into its random (precision) and systematic components (accuracy), and is
provided for every individual pixel.
The main difference between the OMI and TROPOMI algorithms lies in the cloud product that is used to compute air
mass factors. While the QA4ECV OMI product is based on the $O_2$–$O_2$ absorption feature around 477 nm, and considers
a fixed cloud albedo of 0.8 (version 2.0, Veefkind et al., 2016), the TROPOMI product uses the S5P operational cloud
product in CRB (Cloud as Reflecting Boundary) mode (OCRA/ROCINN-CRB; Loyola et al., 2018). The S5P





ROCINN algorithm is based on the $O_2$ A-band around 760 nm and simultaneously retrieves cloud height and cloud
albedo. Systematic differences between the cloud parameters will result in differences in the air mass factors,
influencing the comparisons. To mitigate the impact of this difference between OMI and TROPOMI, we also switch
off the cloud correction by replacing the cloud-corrected AMF by an equivalent clear-sky AMF ($M_{clear}$, no cloud
correction applied) also provided in the L2 product. Based on equation (2-1), the following simple transformation can
be applied:

$$N_{v\_clear} = \frac{M}{M_{clear}} N_v \qquad (2-2)$$

Note that this transformation has an effect on observations with cloud fractions comprised between 0.1 and 0.4. Indeed,
no cloud correction is applied for CF<0.1 and observations with CF>0.4 are filtered out from the analysis.
**2.4    MAX-DOAS datasets**
Multi-axis DOAS (MAX-DOAS) instruments retrieve the abundance of atmospheric trace species in the lowermost
troposphere (Hönninger et al., 2004; Wagner et al., 2004; Wittrock et al., 2004; Heckel et al., 2005). Based on DOAS
analyses (Platt and Stutz, 2008) of the scattered sky light under different viewing elevations, high sensitivity close to
the surface is obtained for the smallest elevation angles, whereas measurements at higher elevations provide
information on the rest of the column. MAX-DOAS measurements have been used in several studies to validate
satellite HCHO columns (Vigouroux et al., 2009; Franco et al., 2015; De Smedt et al., 2015; Chan et al, 2019; 2020;
Ryan et al., 2020; Kumar et al. 2020). However, a global network of MAX-DOAS instruments has not been used yet
for the validation of HCHO columns from space.
Ground-based data used in this study are presented in Table 1. Apart from the QA4ECV MAX-DOAS dataset, which
relies on harmonized HCHO retrievals (Pinardi et al., 2013; QA4ECV D3.8 and D3.9,
http://www.qa4ecv.eu/sites/default/files), the MAX-DOAS data sets used here were generated by instrument principal
investigators using non-harmonised settings. The conversion to vertical columns and/or vertical profiles relies on
methods of various complexity levels. Table 1 includes details about the retrieval strategy adopted by the different
teams. These include:
- • GA: Geometrical approximation, the vertical column is determined using a single-scattering approximation
- adequate for moderately high elevation angles α (typically 30°) so that a simple geometrical air-mass factor
- (AMF≡SCD/VCD=1/sin(α)) (Honninger et al., 2004; Brinksma et al., 2008; Ma et al., 2013) can be used,
- • QA4ECV: the vertical column is calculated using tropospheric AMFs based on climatological profiles and
- aerosol loads as developed during the QA4ECV project (QA4ECV_MAXDOAS_readmefile). These data are
- less sensitive to relative azimuth angle than the purely geometric approximation presented above,
- • OEM: Vertical profile algorithms using an Optimal Estimation Method (Rodgers, 2000): these make use of a-
- priori vertical profiles and associated uncertainties (Friess et al., 2006; Clémer et al 2010; Hendrick et al., 2014;
- Gielen et al., 2017; Wang et al., 2019a; Friedrich et al., 2019; Bösch et al., 2018),
- • PP: Vertical profile algorithms based on parameterized profile shape functions: these make use of analytical
- expressions to represent the trace gas profile using a limited number of parameters (Irie et al., 2009; 2011; Li et
- al., 2010; Vlemmix et al., 2010; Wagner et al., 2011; Beirle et al., 2019).





Both OEM and parameterized profiling approaches provide vertical profiles of aerosols and HCHO with good
sensitivity in the 0-4 km altitude range, in which 1 to 3 independent pieces of information in the vertical dimension
are available (Vlemmix et al., 2015; Friess et al., 2016; 2019). Recent intercomparison studies (Vlemmix et al., 2015;
Friess et al., 2019; Tirpitz et al., 2021) show that both OEM and parameterized inversion approaches lead to consistent
results in terms of tropospheric vertical columns but to larger differences in terms of profiles. The accuracy of the
MAX-DOAS technique depends on the SCD retrieval noise, the uncertainty of the HCHO absorption cross-sections,
the choice of the a-priori profile shape and the uncertainty of the tropospheric AMF calculation. MAX-DOAS HCHO
slant columns from several instruments have been compared during international large-scale campaigns (CINDI-1 and
2, e.g. Pinardi et al., 2013; Kreher et al., 2020) showing relatively large median differences and larger noise compared
to other slant column products comparisons (e.g. $NO_2$). For HCHO, the slant column precision depends strongly on
the signal-to-noise performance of the DOAS instrument with significantly better results for low-noise research-grade
MAX-DOAS instruments (Pinardi et al., 2013; Kreher et al., 2020). The estimated total uncertainty on HCHO VCD
is of the order of 30% to 60% in polluted conditions. This includes both random (~5% to 30% depending on
instrumental signal-to-noise ratio) and systematic (20%) slant column contributions (Pinardi et al., 2013).
**Table 1: MAX-DOAS HCHO datasets included in the validation exercise. GA stands for geometrical approximation, OEM**
**for Optimal Estimation Method and PP for Parametrized Profiling.**

| Station, Country (lat/long) | Owner/ Group | Instrument Type | Retrieval Type | Reference |
|---|---|---|---|---|
| **De Bilt, The Netherlands** (52.10°N, 5.18°E) | KNMI | miniDOAS / Airyx | SCD and VCD from QA4ECV | Vlemmix et al., 2010 QA4ECV |
| **Cabauw, The Netherlands** (51.97°N, 4.93°E) | KNMI | miniDOAS/ Hoffmann | SCD and VCD from QA4ECV | QA4ECV |
| **Uccle, Belgium** (50.78° N, 4.35° E) | BIRA-IASB | Custom-built MAX-DOAS | VCD and profiles from OEM | Dimitropoulou et al, 2020 |
| **Xianghe, China** (39.75° N, 116.96° E) | BIRA-IASB | Custom-built MAX-DOAS | VCD and profiles from OEM | Hendrick et al., 2014; Vlemmix et al., 2015 |
| **Mainz, Germany** (50°N, 8.2°E) | MPIC | Custom-built MAX-DOAS | SCD and VCD from QA4ECV | Wang et al., 2017 QA4ECV |
| **Munich, Germany** (48,13_N, 11.58°E) | LMU | Airyx 2D MAX-DOAS | VCD and profiles from OEM | Chan et al. 2020 |
| **Mohali, India** (30.67°N, 76.74°E) | IISER/MPIC | Custom-built MAX-DOAS | SCD and VCD from QA4ECV | Kumar et al., 2020 QA4ECV |
| **Thessaloniki, Greece** (40.63°N, 22.96°E) | AUTH | Phaethon | SCD and VCD from QA4ECV | Drosoglou et al., 2017 QA4ECV |
| **Madrid, Spain** (40.3°N, 3.7°W) | CSIC | MAX-DOAS | VCD and profiles from OEM | Benavent, et al., 2019. |
| **Fukue, Japan** (36.8°N, 128.7°E) | ChibaU | CHIBA-U MAX-DOAS | VCD and profiles from PP | Irie et al., 2011; 2012; 2015; 2019. |
| **Chiba, Japan** (35.63°N, 140.10°E) | ChibaU | CHIBA-U MAX-DOAS | VCD and profiles from PP | Irie et al., 2011; 2012; 2015; 2019. |
| **Kasuga, Japan** | ChibaU | CHIBA-U MAX- | VCD and profiles from PP | Irie et al., 2011; 2012; 2015; |



| (33.52°N, 130.48°E) | | DOAS | | 2019. |
|---|---|---|---|---|
| **Pantnagar, India** (29°N, 79.47°E) | ChibaU | CHIBA-U MAX-DOAS | VCD and profiles from PP | Irie et al., 2011; 2012; 2015; 2019. |
| **Phimai, Thailand** (15.18°N, 102.56°E) | ChibaU | CHIBA-U MAX-DOAS | VCD and profiles from PP | Irie et al., 2011; 2012; 2015; 2019. |
| **Xianghe, China** (39.75° N, 116.96° E) | USTC | MAX-DOAS | VCD from OEM | |
| **Beijing CAMS, China,** (39.95°N, 116.32°E) | USTC | MAX-DOAS | VCD from GA | |
| **UNAM, Mexico** (19.33°N, 99.18°W) | UNAM | MAX-DOAS | VCD and profiles from OEM Eastwards pointing | Rivera Cardenas et al., 2021 Arellano et al., 2016 |
| **BroadMeadows, Australia** (-37.7°, 144.9°) | Melbourne University ABM | Airyx | VCD from OEM | Ryan et al. 2018; 2020. |

**2.5    Data Use and Method**
For this study, unless specified otherwise, we filter the satellite data based on the quality assurance values (QA)
(Product Readme File). QA>0.5 filters out most observations presenting an error flag or a solar zenith angle larger
than 70°, a cloud radiance fraction (CRF) at 340 nm larger than 0.6, an air mass factor smaller than 0.1, surface
reflectivity larger than 0.2, or an activated snow/ice flag. It should be noted that, in the first versions of the operational
product, the QA values were not correctly assigned over snow/ice regions, above 75° of SZA, and sometimes over
cloudy scenes. This issue has been corrected from version 2.1.3 (July 2020). For this study, we therefore reassigned
QA values using the above-mentioned filters.
We calculated daily gridded data at a resolution of 0.05°x0.05° in latitude/longitude, both for OMI and TROPOMI,
using the Harp atmospheric toolbox. Along the paper, daily and monthly averages are obtained from daily grids. For
each day, we require the region to be filled with a least 50% of valid grid cells, with a minimum of 10 TROPOMI
observations (2 OMI observations).
For the satellite/satellite and the satellite/ground-based comparisons, we calculate the median of the absolute
differences (absolute bias) and the median of the relative differences (relative bias) in each region or station (relative
either to TROPOMI in the case of sat./sat. or to the MAX-DOAS columns in the case of sat./ground-based). The
corresponding median absolute-value deviations (MAD) of the absolute and relative differences are a robust estimate
of the combined observation and comparison variability. The MAD is defined as the median of the absolute-value
deviations from the data's median:

$$MAD = k.median(abs(Diff\_i - median(Diff\_i)))  \tag{2-3}$$

where the factor k=1.4826 is used to ensure a correspondence with the 1-sigma standard deviation for normal
distribution. The bias is considered as statistically significant if it exceeds ErrB=2*MAD/sqrt(N), where N is the
number of collocated pairs (days or months). We also derive correlation, slope and offset of the linear regression using
the robust Teil-Shein estimator (Sen, 1968) as done in Vigouroux et al. (2020).
**3    TROPOMI HCHO tropospheric columns**

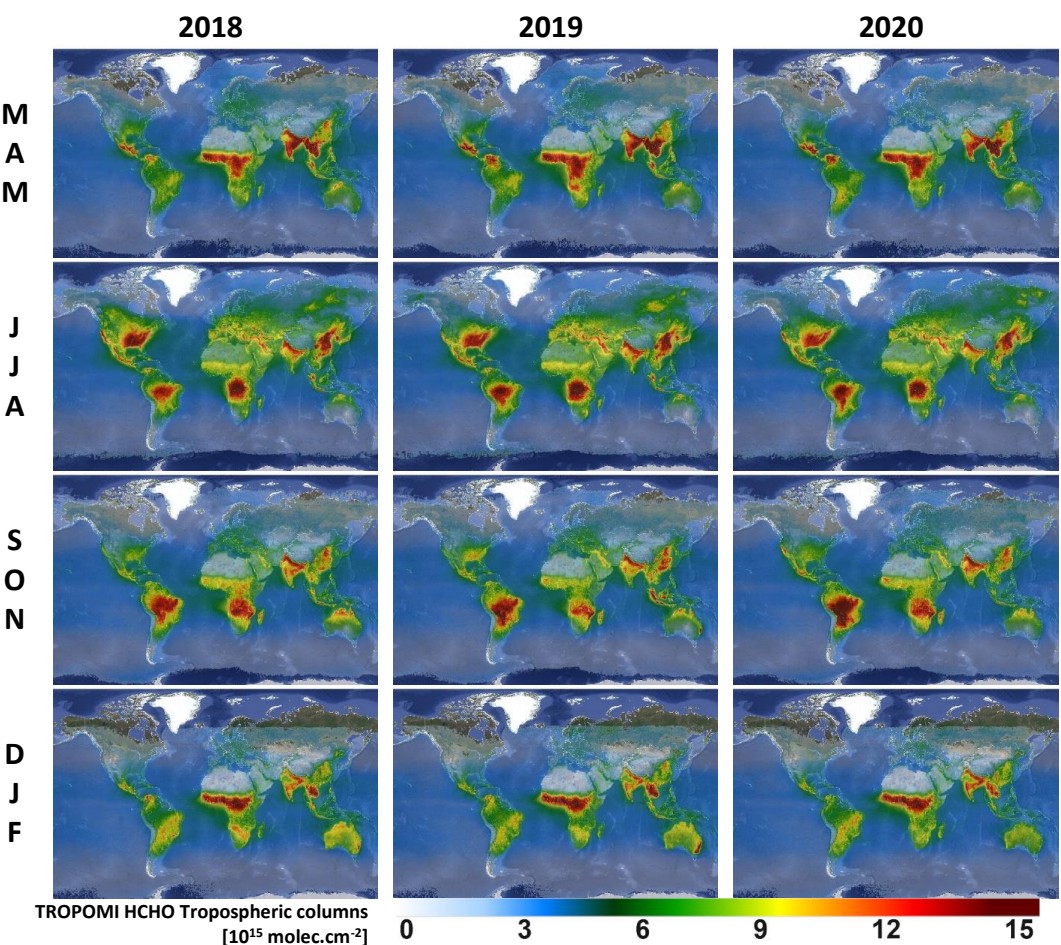

**Figure 1: Seasonal maps of TROPOMI HCHO tropospheric columns during the three first years of measurements (March 2018 – February 2021), on a spatial grid of 0.05° in latitude and longitude. Observations are filtered using the qa_values>0.5. (max.scale: 15x10$^{15}$ molec.cm$^{-2}$). Modified Copernicus Sentinel-5P satellite data, OFFL L2 HCHO product, BIRA-IASB/DLR/ESA/EU.**

As an illustration of the data product, Figure 1 displays the global seasonal distribution of tropospheric HCHO columns
derived from TROPOMI observations between March 2018 and February 2021. The overall seasonality of the HCHO
columns is largely driven by the emissions of NMVOCs from the vegetation and by the interannual variability of
surface temperatures and solar radiation. As can be seen, in South Eastern US for example, the seasonal amplitude is
very important and dominated by biogenic emissions during summertime. On top of biogenic emissions, wildfires
present a large variability. Since 2018, many fire events occurred worldwide and can be traced e.g. in HCHO columns
during summer 2018 and 2020 in Western US, or during summer 2019 in Siberia. After a decrease of about 10 years
(De Smedt et al., 2015), South America experienced two intense fire seasons in 2019 and 2020. The year 2020 was
also marked by the huge Australian and Californian wildfires, respectively, in January and October 2020, detectable





in the seasonal maps. In comparison to biogenic and pyrogenic emissions of natural origin, the contribution due to
anthropogenic NMVOC emissions to the total HCHO columns is generally lower. Although their oxidation is also
enhanced by sunlight, anthropogenic emissions show less seasonality than natural emissions, and their detection is
therefore generally easier in annual maps. This is illustrated in Figure 2, which presents 3-years averages of HCHO
columns over Asia, the Arabic Peninsula, the US and Central and South America, providing detailed information
about the spatial distribution of HCHO at the regional and urban scale. Europe and Africa are shown in the supplement
(fig.S1). Note that the colour scale has been adapted to the regions. Large urban areas are clearly visible in the HCHO
distribution in Asia, the Middle East and South America. With a lower magnitude, US cities are also clearly detectable,
such as Houston, Dallas or Los Angeles. HCHO levels are noticeably lower in Europe, but some urban areas are
visible in the Southern countries.
The quality of the TROPOMI observations also allows observing HCHO columns on a much shorter time scale with
an unprecedented definition. Daily observations of fire plumes are a clear step forward in the satellite remote sensing
of HCHO. They can be observed over much longer distances than before, thanks to the daily global coverage, coupled
with the finer spatial resolution and the improved signal to noise ratio, allowing to detect lower columns transported
further away (Alvarado et al. 2020; Theys et al. 2020). Not only wildfires, but also important anthropogenic emission
plumes can be observed on a daily basis, for example on the Eastern coast of Saudi Arabia. A few illustrations are
given in fig.S2. The TROPOMI performances for the observations of HCHO are discussed more quantitatively along
the paper in terms of precision and bias, as a function of the HCHO levels, and of the temporal and spatial scales.




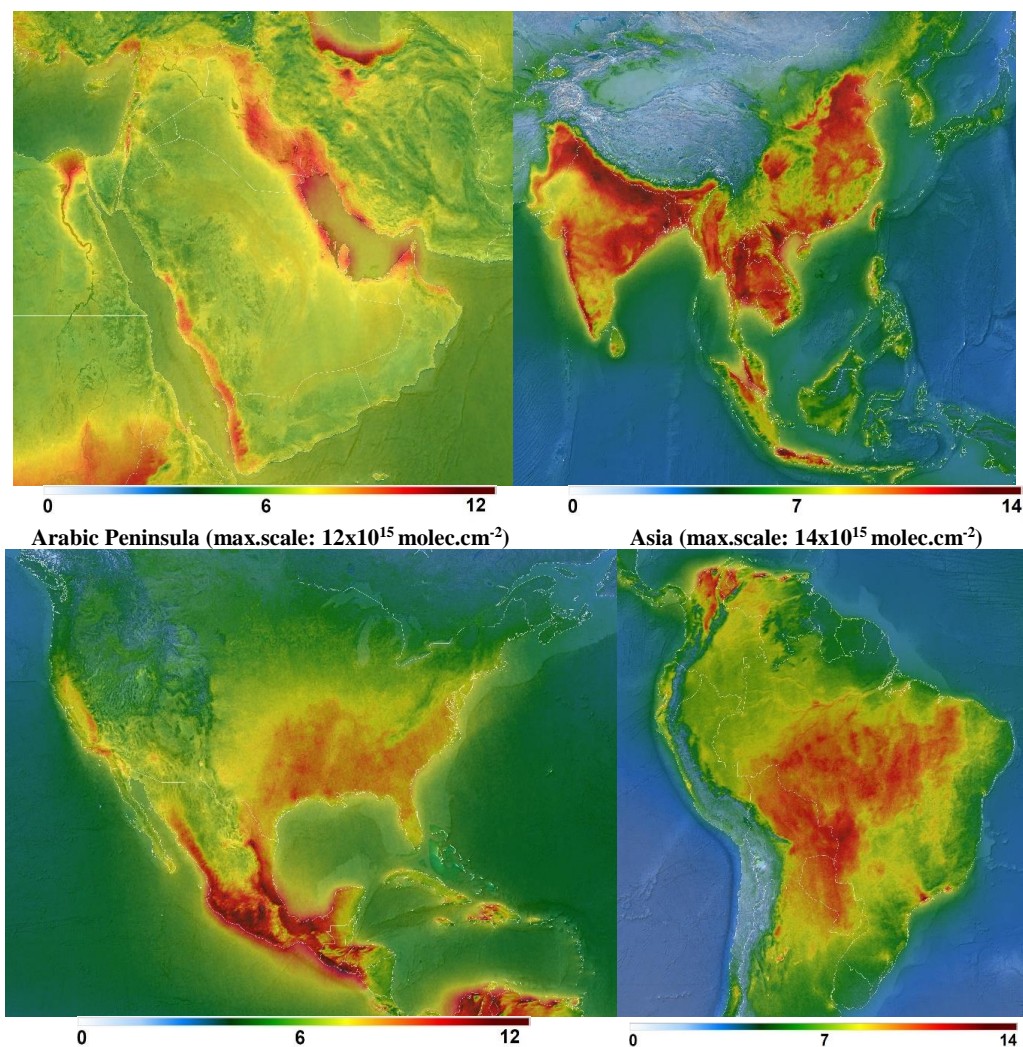

**Figure 2: Multi-annual regional maps of TROPOMI HCHO tropospheric columns (March 2018 – February 2021), on a spatial grid of 0.05° in latitude and longitude. Observations are filtered using the qa_values>0.5. Modified Copernicus Sentinel-5P satellite data, OFFL L2 HCHO product, BIRA-IASB/DLR/ESA/EU.**

## 4    Comparison between OMI and TROPOMI measurements

In this section, we evaluate the consistency between OMI and TROPOMI HCHO tropospheric columns. In addition, we present the gain in precision obtained with TROPOMI. The analysis relies on 32 months of simultaneous measurements from April 2018 to December 2020, allowing for a meaningful comparison at different scales. We first





compare the precision obtained on individual measurements, and then proceed with a comparison of the precisions
achieved when averaging data at different spatial and temporal scales.
**4.1    HCHO slant column precision**
The random uncertainty of the tropospheric HCHO column is dominated by the error on the fitted slant column
densities (SCDE) which is directly related to the signal to noise ratio (SNR) of the measurement.  From this point of
view, TROPOMI performs significantly better than previously launched nadir UV-VIS satellite instruments. In the
spectral range of HCHO retrievals (328.5-359 nm), the SNR of the TROPOMI spectra exceeds pre-flight requirements
that were based on OMI specifications (Kleipool et al., 2018; Ludewig et al., 2020).
Figure 3 presents global maps of SCDE averaged over 3 months during summer 2019, from OMI and TROPOMI.
From the improved SNR of TROPOMI in the UV range, TROPOMI HCHO SCDEs of individual observations are
about 25% lower than OMI ones. Over remote areas, the TROPOMI SCDE is about $6\times10^{15}$ molec.cm$^{-2}$, while it is
$8\times10^{15}$ molec.cm$^{-2}$ for OMI. Slant column density errors are also improved over emission areas and at larger SZA.
Contrary to OMI, the effect of the South Atlantic Anomaly is absent in TROPOMI SCDE. This probably results from
a better shielding of the instrument against extra-terrestrial high energy radiation. The implemented iterative spike
algorithm (De Smedt et al., 2018) is also more efficient because of the lower noise level of the instrument. Note
however that over mountains, TROPOMI SCDE are higher than OMI ones. The most obvious effect is observed over
the Himalayans, but other chains such as the Andes or the Rocky mountains are also affected. This effect has been
identified as a scene inhomogeneity effect (Richter et al., 2018; 2020). The effect is also visible along the borders of
bright lakes or white surfaces. OMI retrievals are also affected by scene inhomogeneity effects, but the larger size of
the ground pixels and the larger mean SCDE values make its detection more difficult. We note that in the long-term
averaged maps of the HCHO tropospheric columns, some collocated artefacts appear (Figure 2, e.g. the white sands
in the US, Tuz Golu lake in Turkey or Lake Mackay in Australia). Most of the snow/ice scenes are eliminated by the
quality assurance values. The observations could however be better filtered over mountains and along the lake borders,
or even corrected during the fit of the slant columns as demonstrated for $NO_2$ and glyoxal (Lerot et al., 2021, in prep.).
The relatively coarse albedo climatology also needs to be updated with a TROPOMI-based product, better defined in
space and time (Loyola et al., 2020).

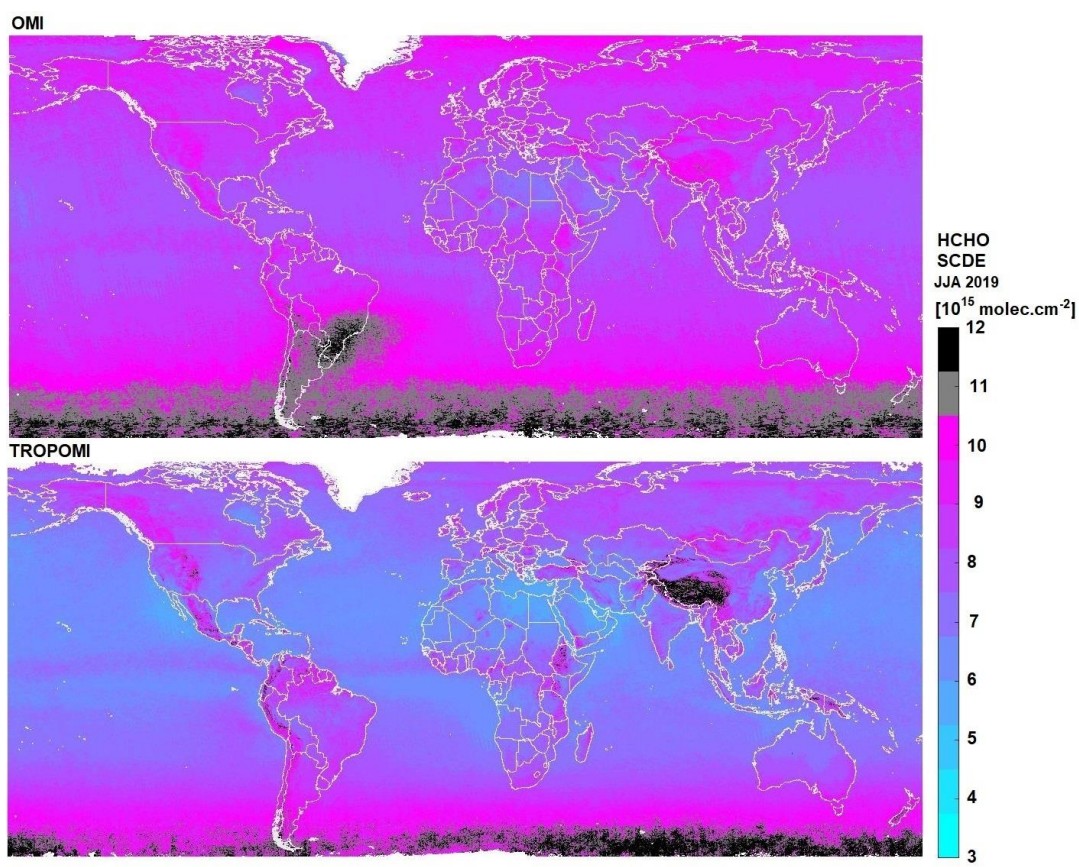


**Figure 3: Average HCHO slant column density fitting error (SCDE) retrieved from OMI (upper panel) and TROPOMI (lower panel) in JJA 2019, on a spatial grid of 0.05° in latitude and longitude.**

The OMI SCDEs have been very stable over the years, showing a limited increase of about 5% between 2005 and 2019 (De Smedt et al., 2018). However, the number of valid OMI observations has decreased by about 30% during the same period (-50% at large SZA) due to the row anomaly. In order to evaluate the stability of the TROPOMI HCHO retrievals during the three first years, Figure 4 presents the time series of the TROPOMI HCHO slant column errors in the remote Pacific Ocean as a function of latitude and instrumental rows. As expected, we observe an increase of the noise for large SZAs, and for the 25 first and last rows of the scan, which have a different detector binning (L1b ATBD). The fact that the algorithm makes use of daily updated radiances as reference for the DOAS fit allows for very stable results in time and across the rows. Only the change in pixel size in August 2019 (L1b readme file) resulted in a moderate step increase of the SCDE of about 15%. These values are compared to the observed standard deviation of the slant columns in the same regions (see fig.S3). We observe a very good agreement between the SCDEs and the standard deviation, indicating that they give a good representation of the random errors.

The reported uncertainty on the tropospheric vertical columns due to random errors corresponds to the SCDE divided by the AMF for each observation. In the Equatorial Pacific, the TROPOMI vertical column precision is about $5 \times 10^{15}$



molec.cm$^{-2}$, while it is 7x10$^{15}$ molec.cm$^{-2}$ for OMI. It is larger over continental emissions, where the AMFs are
generally smaller than 1.

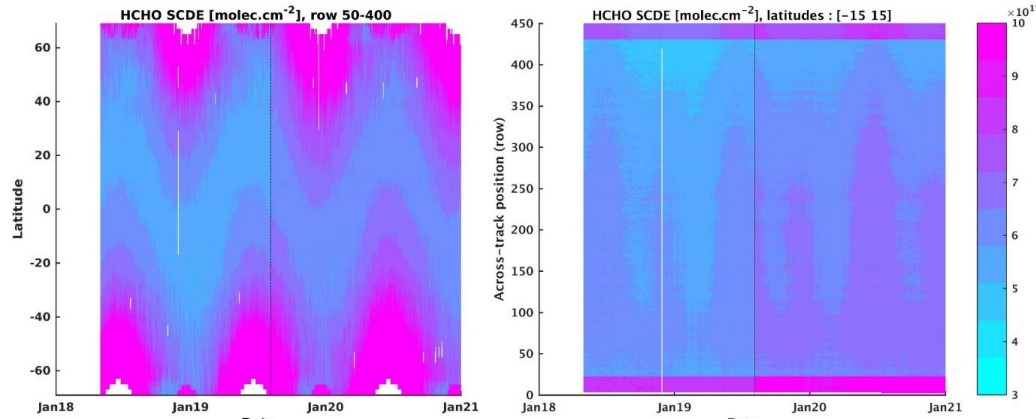

**Figure 4: TROPOMI HCHO slant column density errors (SCDE) as a function of the latitude (left column) or the detector**
**row (right column). The step increase on 6th August 2019 reflects the change in the TROPOMI pixel size (indicated with**
**the black line).**
**4.2    HCHO tropospheric columns**
Figure 5 presents the yearly averaged OMI and TROPOMI HCHO vertical columns ($N_{v\_clear}$) for 2019. Even at this
level of averaging, the lower noise level of TROPOMI is very clear, especially for low to medium HCHO levels. We
observe an overall good agreement of the columns both in magnitude and in their spatial distribution. Differences of
TROPOMI and OMI yearly averages range from +2x10$^{15}$ molec.cm$^{-2}$ over Tropics to -2x10$^{15}$ molec.cm$^{-2}$ over mid-
latitude regions. Differences tend to increase with latitudes. However, as the quality of the TROPOMI observations is
improved at large solar zenith angles, more data in winter months are kept in the TROPOMI dataset, which can
influence yearly averaged columns at those latitudes. In order to provide quantitative comparisons, we calculated daily
and monthly averaged columns in 35 regions covering a broad range of emission levels and observation conditions
(large black boxes on Figure 5). As the regions are large, many observations are included (on average 500/day for
OMI, 12500/day for TROPOMI). To obtain daily and monthly comparison pairs, we keep coincident days of
observations and follow the methodology presented in sect. 2.5.




**Figure 5: Average HCHO tropospheric column ($N_{v\_clear}$) retrieved from OMI (first line) and TROPOMI (second line) in**
**2019. Limits of the regions selected for the comparisons are shown on the TROPOMI map. Differences between OMI and**
**TROPOMI maps are shown on the last panel. The same grid is used for both dataset (0.05°). Data are filtered using the**
**product quality flags. The large black boxes on the TROPOMI maps represent the regions used in the comparisons (see**
**Figure 6 and Figure 7).**





An example of a time series over Equatorial Africa is presented on the first panel of Figure 6, where monthly averaged
$N_{v\_clear}$ are shown, and comparison numbers are provided in the inset. In the Equatorial African region, the seasonal
cycle is marked by two peaks during the dry seasons and two minima during the wet seasons. In 2019, the minimum
was particularly low, observed in both the OMI and TROPOMI timeseries, while the maxima tend to increase over
the years. More examples of time series can be found in fig.S4. In all the regions, the seasonal and interannual
variability of the HCHO columns are observed very consistently with OMI and TROPOMI.

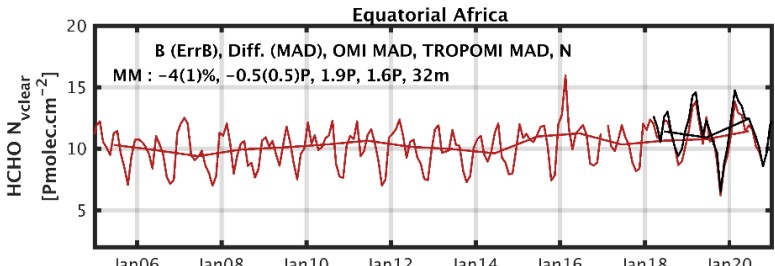

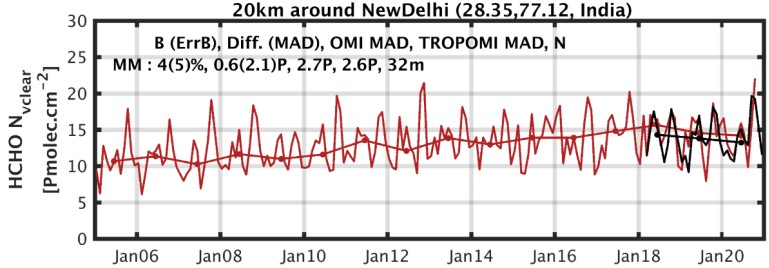

**Figure 6: Examples of monthly and yearly averaged HCHO columns ($N_{v\_clear}$) retrieved from OMI (Oct.2004-Dec.2020, in**
**red) and TROPOMI (2018-Dec.2020, in black) at two different spatial scales selected for the comparison: a large region of**
**Equatorial Africa, and a circle of 20km-radius over New Delhi in India. Absolute and relative biases between OMI and**
**TROPOMI HCHO monthly averaged columns are given in inset, as well as the median deviations of the OMI and**
**TROPOMI averaged columns. [Pmolec.cm$^{-2}$ = 1x10$^{15}$ molec.cm$^{-2}$].**
Figure 7 presents the absolute and relative biases between OMI and TROPOMI HCHO tropospheric columns for all
regions. Numbers are provided for daily averaged columns applying a cloud correction ($N_v$) or not ($N_{v\_clear}$). Regions
are sorted as a function of the averaged TROPOMI HCHO column. At this large spatial scale, the regions over
Equatorial Africa, Northern China and Northern India present the largest annual columns worldwide, with median
levels larger than $10 \times 10^{15}$ molec.cm$^{-2}$. Tropical regions in South America, Africa and Asia present elevated levels of
HCHO as well, with annual averaged columns larger than $8 \times 10^{15}$ molec.cm$^{-2}$.
Looking at $N_v$ comparisons, it appears that the OMI HCHO columns present a positive bias compared to TROPOMI
from $17 \pm 2.5\%$ for the columns larger than $5 \times 10^{15}$ molec.cm$^{-2}$, to $30 \pm 5\%$ for the lower columns. This bias exceeds
50% in Northern latitudes ($>45°$) and low-emissions ($<2 \times 10^{15}$ molec.cm$^{-2}$) regions of Canada and Alaska. However,
when comparing $N_{v\_clear}$, the biases are strongly reduced below 10% in all regions where the HCHO levels are larger
than $5 \times 10^{15}$ molec.cm$^{-2}$, and the TROPOMI columns are found to be slightly larger than OMI on average ($-3 \pm 1.2\%$).
In mid-Northern-latitudes/moderate emissions ($2-5 \times 10^{15}$ molec.cm$^{-2}$) regions such as Europe, Central and Western



US, North Western Canada, Siberia or Tibet, OMI columns present a remaining bias of about 15±3%, while in the
regions of Canada and Alaska, a larger bias of about +30±7% remains. Note that we observe biases lower than 10%
in the Maghreb and Southern Australia regions, despite their relatively low columns or low latitudes.
We conclude that biases up to 30% related to the cloud correction are observed over Tropical regions where the clouds
are the highest in altitude (Africa, South America, South Asia), and a smaller but systematic effect, up to 15%, is
observed over mid-latitude polluted regions such as China, India, US or Europe. We also note that the differences
between $N_v$ and $N_{v\_clear}$ are mainly significant for the OMI HCHO columns. It has been reported that the cloud
pressures retrieved from TROPOMI and from OMI present a bias (OMI clouds are higher in altitude, Compernolle et
al., 2020). This translates into OMI cloud-corrected air mass factors generally smaller than TROPOMI AMFs by 5 to
30%, depending on the cloud altitude, and therefore in a positive bias of the OMI HCHO VCD compared to the
TROPOMI product. It is therefore important to keep in mind that the use of different cloud products may introduce
inconsistencies, which may be resolved by using clear HCHO VCDs ($N_{v\_clear}$).
Figure 8 shows the linear regression between OMI and TROPOMI monthly averaged columns, considering all regions
together. The relation between OMI and TROPOMI is provided for $N_v$ and $N_{v\_clear}$. This shows that switching off the
cloud correction in the OMI and TROPOMI HCHO products allows to significantly improve not only the slope (from
0.87 to 0.92) and the intercept (from 1.52 to $0.48 \times 10^{15}$ molec.cm$^{-2}$), but also the data scatter, i.e. the Pearson R
correlation (from 0.74 to 0.98). When considering large-scale comparisons, the agreement between OMI and
TROPOMI $N_{v\_clear}$ is therefore very satisfactory.

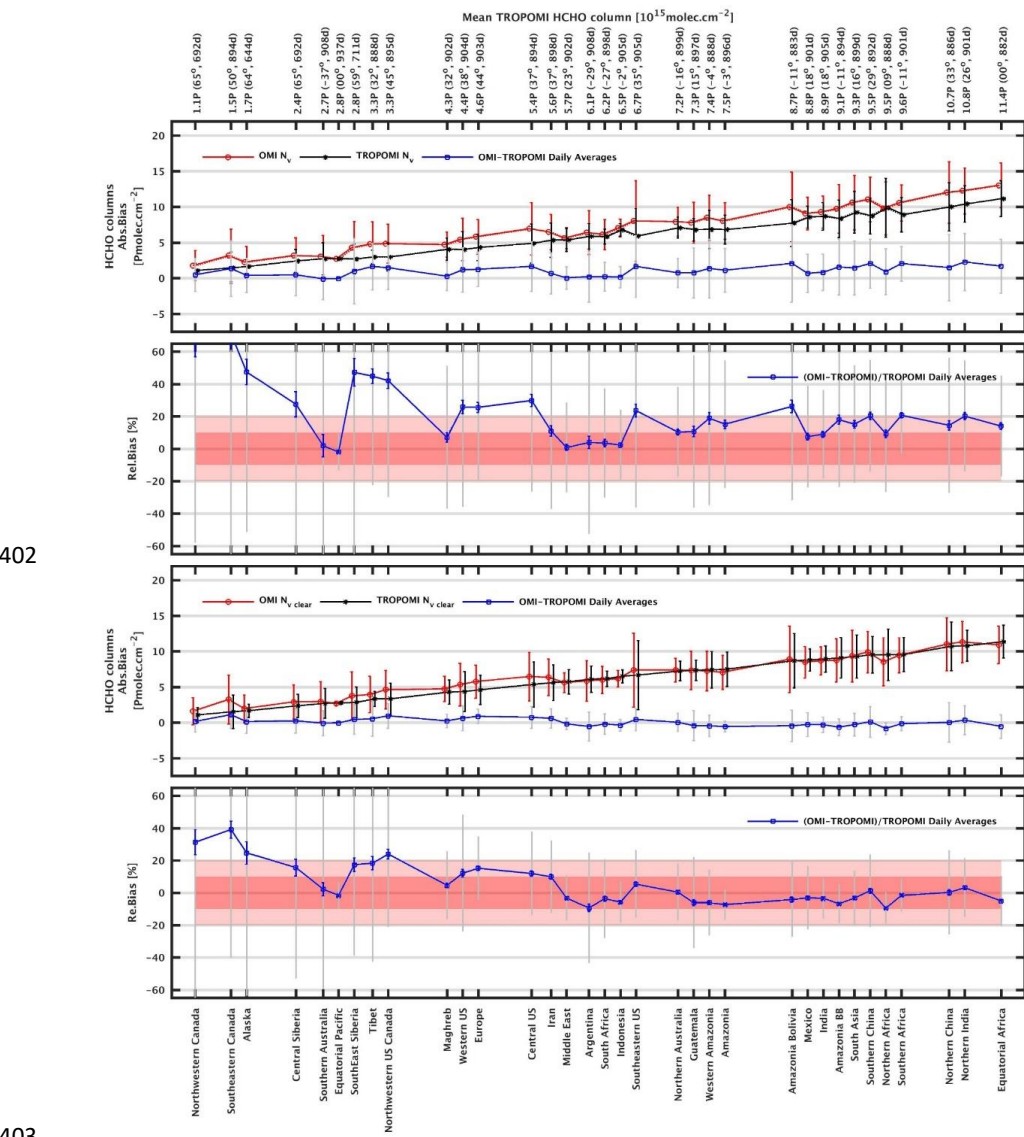

**Figure 7: Absolute and relative biases between OMI and TROPOMI HCHO daily averaged tropospheric columns using cloud corrected AMF ($N_v$, two upper panels) or clear sky AMF ($N_{v\_clear}$, two bottom panels) for the large regions represented on Figure 5. Regions are sorted as a function of the median TROPOMI HCHO column. Values of the averaged HCHO columns are provided on the top axis, as well as the numbers of common days taken for the comparison and the latitude of the region. The median OMI (red) and TROPOMI (black) columns are plotted together with the absolute differences (in blue). Error bars represent the median deviations of the columns, or the median absolute deviations of the**





**differences (MAD, in grey). Statistical ErrB are also plotted for the relative bias (in blue). Pink areas indicate 10% and**
**20% bias. [Pmolec.cm$^{-2}$ = 1x10$^{15}$ molec.cm$^{-2}$].**

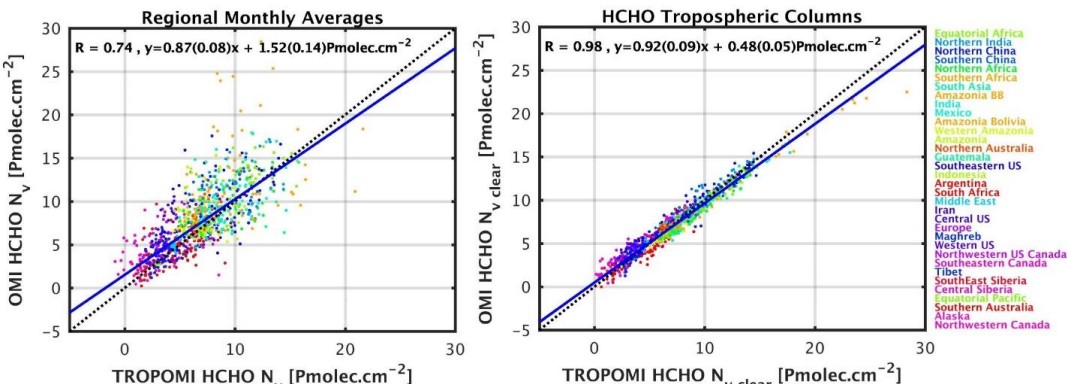


**Figure 8: Scatter plots of OMI versus TROPOMI columns for the monthly means of collocated data. Results are shown for**
**$N_v$ (left panel) and $N_{v\_clear}$ (right panel). The correlation, slope and intercept of a linear regression using the robust Teil-**
**Shein estimator are given as inset and plotted as a blue line. Black dotted line is the 1:1 line. The color indicates the latitude**
**of the region. [Pmolec.cm$^{-2}$ = 1x10$^{15}$ molec.cm$^{-2}$].**
When averaging data over large regions, the dispersion due to random uncertainties is greatly reduced compared to
individual observations. As summarized in Table 2, the median absolute deviations of the monthly averaged columns
are equivalent for OMI and TROPOMI (1.8x10$^{15}$ molec.cm$^{-2}$), while the MAD of their differences are significantly
lower (0.5x10$^{15}$ molec.cm$^{-2}$). This indicates that at this spatiotemporal resolution, the natural variability dominates the
dispersion of the averaged observations. Looking at the daily averaged columns, the TROPOMI median deviation is
lower than for OMI (2.2/2.7), but still larger than the MAD of their differences (1.5).
The improved spatial resolution of TROPOMI should allow for a better detection of localized HCHO emissions. To
address this question, we performed the same comparisons as for the large regions, but looking at smaller areas of
20km radius around cities. Figure 9 presents the absolute and relative biases of the monthly averaged HCHO columns
($N_{v\_clear}$) for a large number of cities. At this spatial scale, Jakarta is the location with the largest median HCHO level
(>18x10$^{15}$ molec.cm$^{-2}$ over the 2018-2020 period). Indian, Chinese and other Asian cities follow, as well as Mexico,
Monterrey or Kinshasa (>12x10$^{15}$ molec.cm$^{-2}$). Sao Paulo, Tehran and Cairo present also noticeably elevated HCHO
levels (>9x10$^{15}$ molec.cm$^{-2}$). An example over New Delhi is presented on the second panel of Figure 6 and more
examples can be found in fig.S5.
When comparing OMI and TROPOMI $N_{v\_clear}$ around the cities, the same general behaviour as in the large regions
can be observed. OMI presents a positive bias (20±15%) compared to TROPOMI for low to medium HCHO levels,
while for medium to large levels, the agreement is very good on average (-1±10%). There are nevertheless a few
exceptions where TROPOMI HCHO columns are significantly larger than the OMI ones. This is the case at La
Reunion, Paramaribo, Nairobi, Bujumbura, Sao Paulo, Monterrey, Mexico, or Jakarta. Those cities are located along
marine coasts or lakes, at higher altitude, or are surrounded by mountains. In those cases, the finer spatial resolution
of TROPOMI clearly improves the detection of the HCHO signal. For most other locations, however, the impact of
the improved spatial resolution of TROPOMI on the HCHO columns is not detectable in the column magnitudes,





when compared to OMI observations. This is likely related to the nature of the HCHO production that mostly is
secondary from the oxidation of NMVOCs with various lifetimes (Stavrakou et al. 2015; Bauwens et al., 2016). Except
for regions where the topography presents sharp discontinuities, this causes a natural spread of the HCHO columns at
a scale larger than the TROPOMI spatial resolution.
Note however that at this spatial resolution (20km radius), the level of noise is larger than for the regional averages
and the TROPOMI averaged columns are significantly more stable than the OMI ones, as evidenced by their median
deviations (see ). On a daily basis, the OMI columns present a dispersion of $7.8 \times 10^{15}$ molec.cm$^{-2}$, while the TROPOMI
dispersion is about twice smaller ($3.7 \times 10^{15}$ molec.cm$^{-2}$). In this case, the MAD of the differences ($7.1 \times 10^{15}$ molec.cm$^{-2}$
$^{2}$) is dominated by the noise on OMI observations. Note that these estimates still include the natural variability of the
columns themselves. If an area of 20-km in the remote Equatorial Pacific is considered, the observations represent
constant background values and the seasonal variability is further reduced. In such conditions, the dispersion of the
OMI daily observations is $3.5 \times 10^{15}$ molec.cm$^{-2}$, while only $1 \times 10^{15}$ molec.cm$^{-2}$ for TROPOMI. We show in the next
section that validation with ground-based measurements brings further information on the satellite column precision.

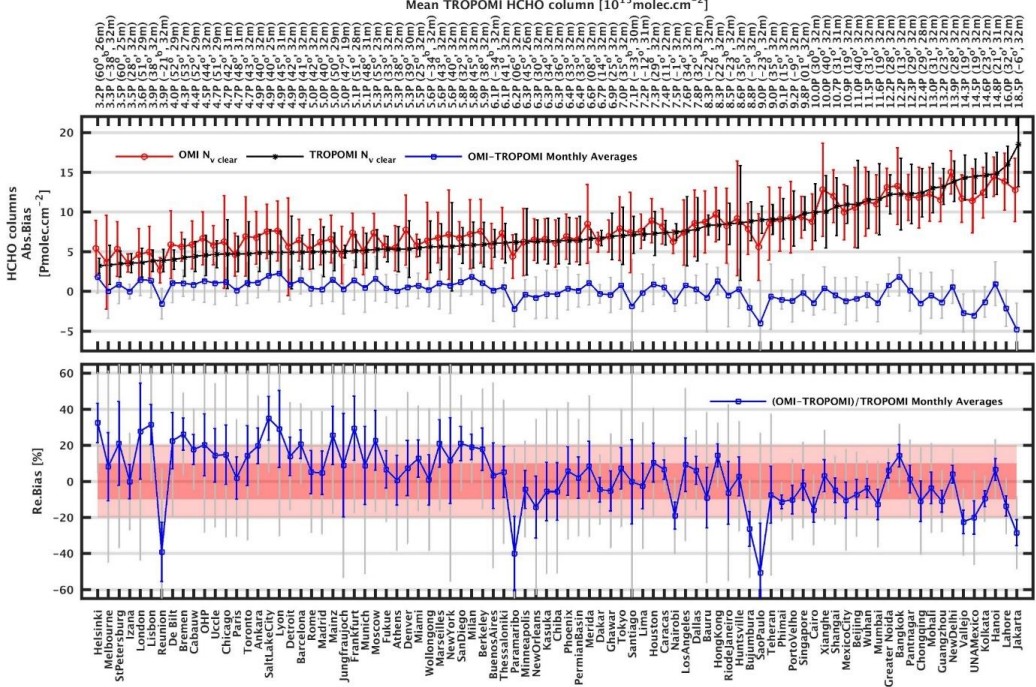


**Figure 9: Absolute and relative biases between OMI and TROPOMI HCHO monthly averaged tropospheric columns using**
**clear sky AMF ($N_{v\_clear}$) within 20km-radius circles around selected cities, sorted as a function of the median TROPOMI**
**HCHO column. Value of the averaged HCHO columns are provided on the top axis, as well as the numbers of months taken**
**for the comparison, and the latitude of the region. The median OMI (red) and TROPOMI (black) columns are plotted**
**together with the absolute differences (in blue). Error bars represent the median absolute deviations (MAD) of the columns**
**and of the differences (in grey). Statistical ErrB are also plotted for the relative bias (in blue). Pink areas indicate 10% and**
**20% bias. [Pmolec.cm$^{-2}$ = $1 \times 10^{15}$ molec.cm$^{-2}$].**





**Table 2: Median absolute deviation of the OMI and TROPOMI daily and monthly averaged columns ($N_{v\_clear}$), in large regions and in 20km-radius area. MAD of differences between OMI and TROPOMI columns are also given in the last column.**

| Dispersion | OMI MAD [$10^{15}$ molec.cm$^{-2}$] | TROPOMI MAD [$10^{15}$ molec.cm$^{-2}$] | OMI-TROPOMI MAD [$10^{15}$ molec.cm$^{-2}$] |
|---|---|---|---|
| Monthly Regional | 1.8 | 1.8 | 0.5 |
| Daily Regional | 2.7 | 2.2 | 1.6 |
| Monthly 20km | 3.3 | 2.5 | 2.4 |
| Daily 20km | 7.8 | 3.7 | 7.1 |
| Daily 20km in the Equatorial Pacific | 3.5 | 1.0 | 3.7 |

## 5 Validation with a global MAX-DOAS network

Here, we present a validation exercise based on a network of 18 ground-based MAX-DOAS instruments. This effort complements the study of Vigouroux et al. (2020), which relied on a network of FTIR instruments. Compared to the FTIR instruments, the MAX-DOAS provide a higher sensitivity in the boundary layer, where the bulk of HCHO is located. The MAX-DOAS network covers stations where the level of HCHO is significant, from medium to very large HCHO columns, while the FTIR network includes a larger number of remote stations. In this study, we validate in parallel the OMI and TROPOMI datasets. We first focus on a direct comparison of the satellite and MAX-DOAS tropospheric columns. The effect of the vertical smoothing is investigated in the next subsection for three stations.

### 5.1 Direct comparisons of tropospheric columns

For each station in Table 1, we consider daily averages of the satellite columns in a radius of 20km around the instruments. We average MAX-DOAS columns between 11h and 16h local time. We keep coincident days of observations (OMI/MAX-DOAS, TROPOMI/MAX-DOAS) to obtain daily and monthly comparison pairs. Note that the time periods used for the comparison are not the same for OMI and TROPOMI, and vary between the stations. To obtain the validation results, we follow the methodology presented in Vigouroux et al. (2020) (see sect. 2.5).

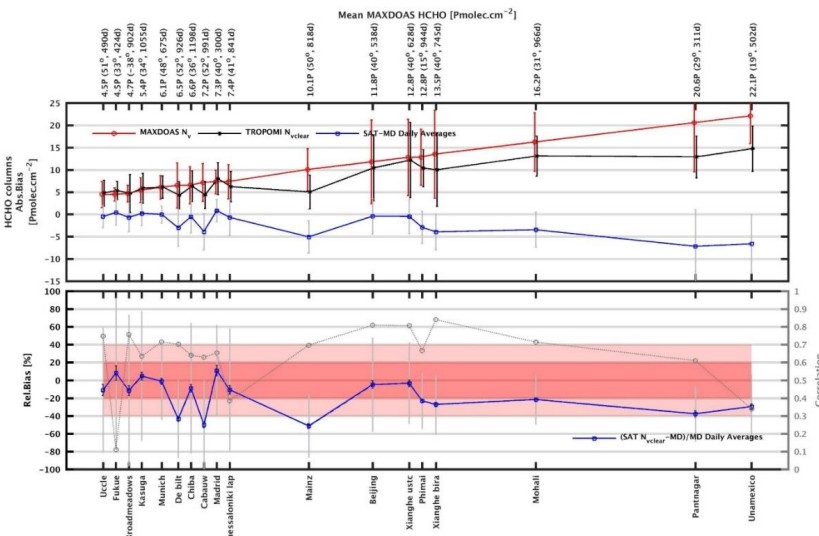

**Figure 10: Absolute (top, blue line) and relative biases (bottom) between MAX-DOAS and TROPOMI HCHO daily averaged tropospheric columns in a circle of 20km-radius around the stations. Regions are sorted as a function of the median MAX-DOAS HCHO column. In the upper plot, the median MAX-DOAS (red) and TROPOMI (black) columns are plotted together with the differences. Error bars (in grey) represent the median absolute deviations (MAD) of the columns and of the differences. Statistical ErrB are also plotted for the relative bias (in blue). Pink areas indicate 20% and 40% bias. The correlation between the daily observations are given in the lower plot (grey circles). [Pmolec.cm$^{-2}$ = 1x10$^{15}$ molec.cm-$^{2}$].**

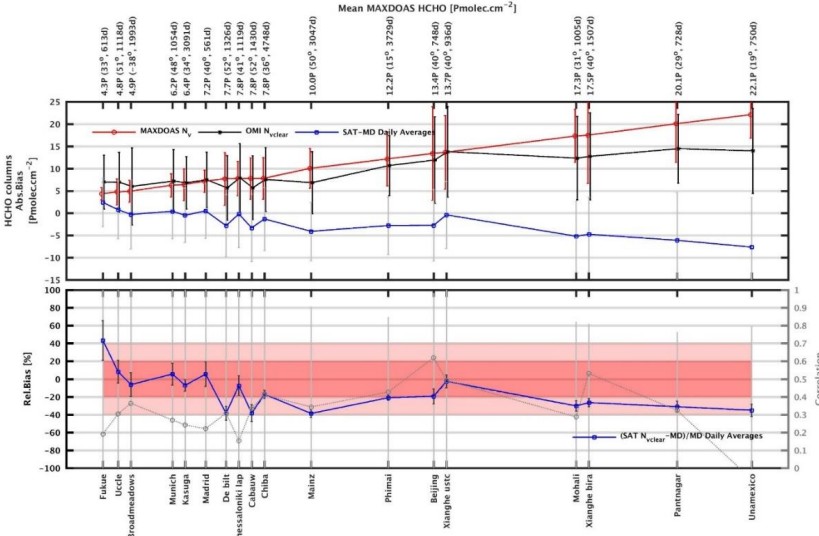

**Figure 11: same as Figure 10 for MAX-DOAS and OMI HCHO daily averaged.**



Figure 10 and Figure 11 present the absolute and relative biases of the daily averaged columns as a function of the
median MAX-DOAS HCHO column, respectively, for TROPOMI and OMI. A more detailed description for each
station and for individual time series is presented afterwards.  The values of the biases are similar for OMI and
TROPOMI, except for the lowest columns in Uccle and Fukue, where OMI presents larger positive biases exceeding
+20%. In agreement with Vigouroux et al. (2020), TROPOMI columns do not present a significant bias for the range
of HCHO levels from 4 to $8x10^{15}$ molec.cm$^{-2}$. Note that, in contrast to FTIR data, the range of values covered by our
MAX-DOAS network does not extend to columns lower than $4x10^{15}$ molec.cm$^{-2}$. We observe that the stations in De
Bilt and Cabauw tend to show somewhat stronger negative biases even for medium levels of HCHO, which might
point to a network inhomogeneity.  For larger HCHO columns ($>8x10^{15}$ molec.cm$^{-2}$), and in agreement with the FTIR
results, we observe that negative biases tend to increase for large HCHO columns such that the underestimation of the
satellite columns reaches about -40% for the largest columns. On the upper plot, the error bars represent the median
absolute deviations of the columns and of their differences. It appears clearly that the MADs obtained with TROPOMI
are substantially lower than those obtained with OMI. Note that the type of MAX-DOAS instrument (in particular its
signal-to-noise ratio) may also influence the observed MAD at the different stations.
Figure 12, Figure 13 and Figure 14 present more detailed results for the stations in Europe, Japan and Australia, and
China, India, Thailand and Mexico. On each plot, the time series of the MAX-DOAS, OMI and TROPOMI data are
displayed together. Results of the daily statistical analysis are given as inset. At European stations, which show
medium range HCHO levels, we obtain contrasted results. With a mean HCHO column of $4.5x10^{15}$ molec.cm$^{-2}$, Uccle
is one of the stations with the lowest columns of the network presented in this paper. While OMI values show a
positive bias (13±15%) and a poor correlation (0.3) with the MAX-DOAS, TROPOMI appears to be biased low (-
10±6%) but much better correlated (0.82) with the MAX-DOAS data. As opposed to Uccle, the observed biases in De
Bilt, Cabauw, and Mainz are largely negative (from -40% to -50%). The correlations found with TROPOMI are
nevertheless much better than with OMI. Note that the median MAX-DOAS HCHO value in Mainz is larger than
$10x10^{15}$ molec.cm$^{-2}$, which is quite high for an European site. The results in Munich have been presented in details in
Chan et al. (2020). They are closer to what is found in Uccle, with a small positive bias for TROPOMI (1±3%) and
for OMI (6±13%). Similarly in Madrid, OMI and TROPOMI results are very consistent with a mean bias of
respectively 8±16% and 10±6%. In Thessaloniki, the negative bias is -12±5%, but the correlation is poorer than in
Madrid.



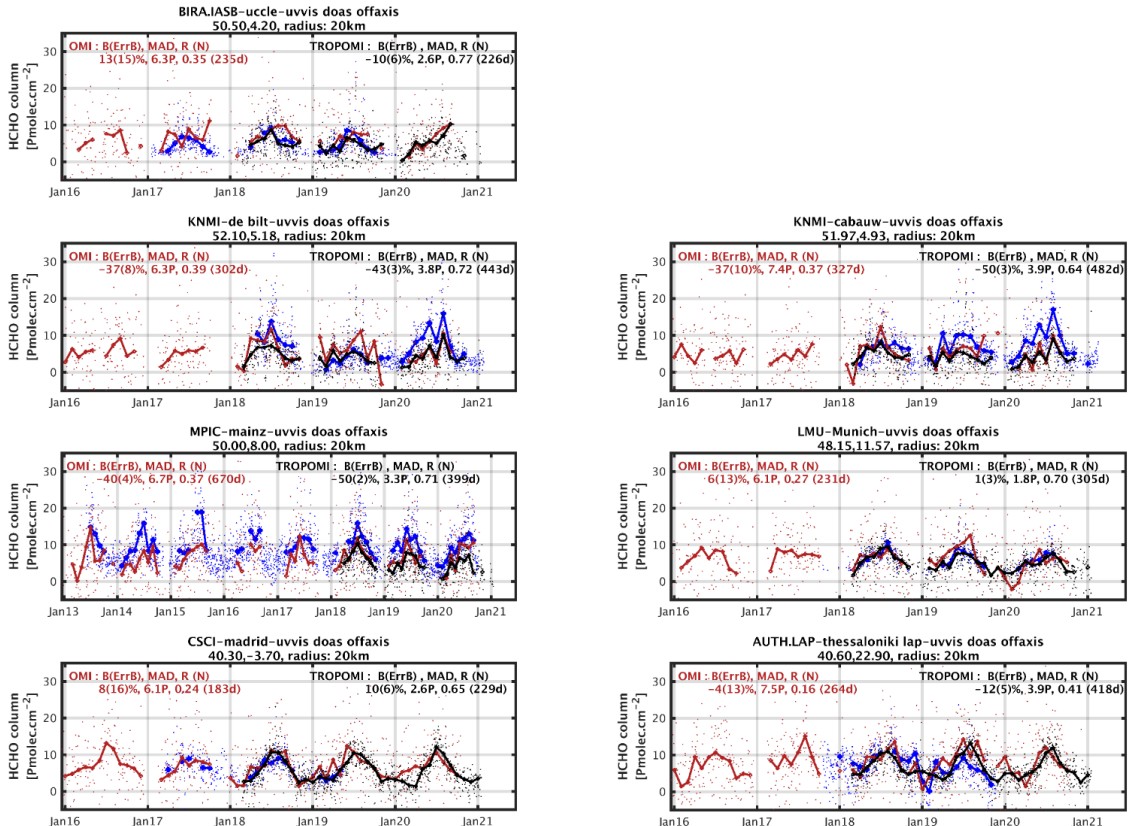

**Figure 12: Time series of MAX-DOAS HCHO columns (blue), OMI $N_{v\_clear}$ (red) and TROPOMI $N_{v\_clear}$ (black) at European sites. Thick lines show monthly median values and dots represent daily median values. Mean relative bias, median absolute deviations and correlations between the time series are provided for the daily averaged data. [Pmolec.cm$^{-2}$=10$^{15}$ molec.cm$^{-2}$].**

In Figure 13, we show three Japanese stations operated by the CHIBA University. Mean HCHO levels in Japan are comparable to values found at European sites. In Chiba and Kasuga, TROPOMI and MAX-DOAS columns are strongly correlated (about 0.7), but on the island of Fukue the correlation is poor due to a lack of variability at this site. At all these sites, TROPOMI shows small biases compared to MAX-DOAS data (-9±4% in Chiba, 3±4% in Kasuga, 8±8% in Fukue). The HCHO observations in Broadmeadows, in Northern Melbourne, have been published by Ryan et al. (2020). We find a bias of -12±6% for TROPOMI and a good correlation of about 0.7. Quite unusually, the seasonal amplitude of the MAX-DOAS time series at this station is smaller than observed with OMI and TROPOMI.



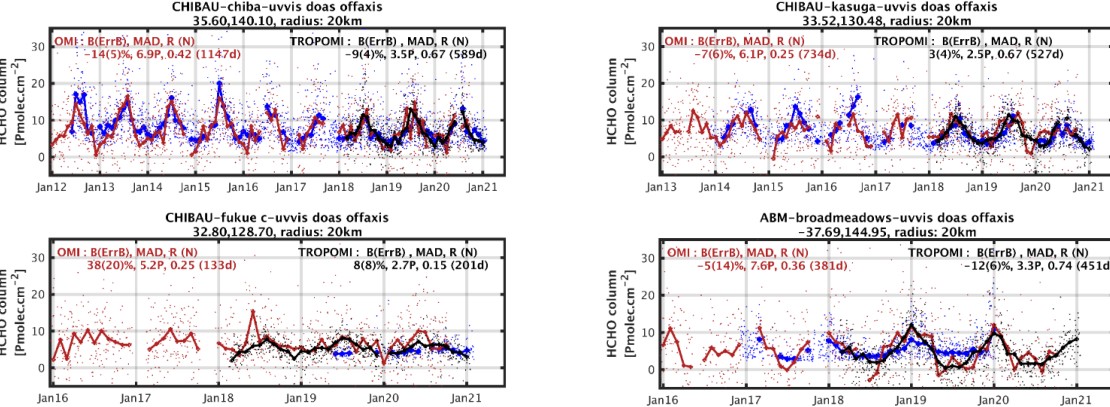

**Figure 13: Same as Figure 12 in Japan and Australia.**

Stations with large HCHO levels in China, India, Thailand and Mexico are presented in Figure 14. In China, we show the results of two instruments in Xianghe, and one instrument in Beijing. With the USTC instruments, we find small biases of -4±4% and -5±5% and correlations larger than 0.8. With the BIRA-IASB instrument in Xianghe, the correlation is also excellent. The MAX-DOAS columns are larger than the ones obtained with the USTC instrument, and we find a significant negative bias of the TROPOMI data of -27±2%. This result illustrates the actual uncertainty related to the ground-based measurements themselves and the need for further harmonisation of the MAX-DOAS network. Correlations in India and Thailand are of about 0.7, while the biases are consistently negative (-21±2% in Mohali, -38±4% in Pantnagar, -21±2% in Phimae). The situation is more complex at the UNAM site in Mexico. There, the correlation is poor (0.3), and a negative bias of -29±3% is found. These results are however more dependent on the radius considered around the station, and on the selection of the MAX-DOAS observations (Rivera Cárdenas et al., 2021) (see sect. 5.4).

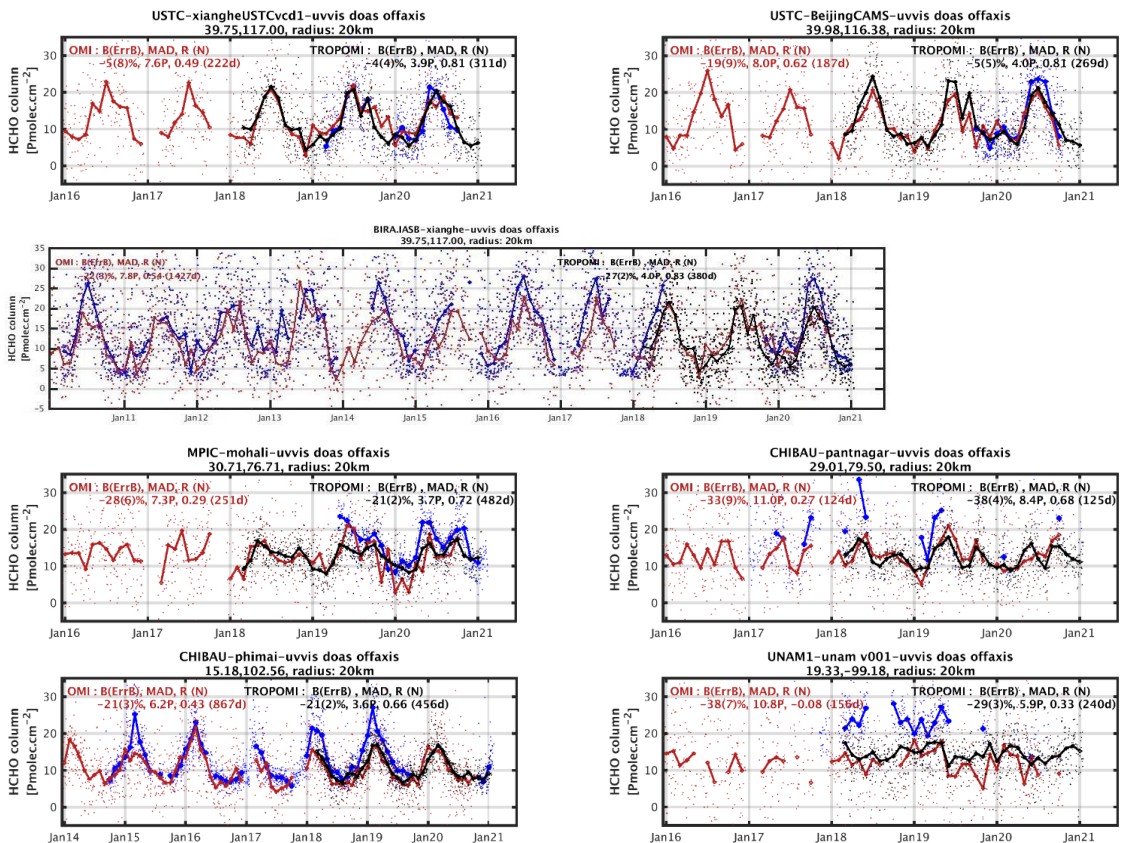

**Figure 14: Same as Figure 12 at Chinese, Indian, Thailand and Mexican sites.**
Finally, Figure 15 presents scatter plots of the satellite against MAX-DOAS columns, considering all the stations and
for daily and monthly comparisons. Table 3 summarizes the validation results. The best agreement is found with
monthly TROPOMI columns, for which we find a slope of 0.64 and a positive offset of $1.7 \times 10^{15}$ molec.cm$^{-2}$ compared
to the MAX-DOAS columns. Slopes and biases for the large columns are found to be close for OMI and TROPOMI
datasets. The improvement with TROPOMI can be seen in the correlation, offset, and bias values obtained for the
lower columns, as well as in the precision of the daily validation results. On average, the OMI biases are found to be
statistically non-significant for the lowest columns. When considering monthly averaged data, the correlation between
MAX-DOAS and satellite columns improves from 0.74 with OMI to 0.85 with TROPOMI (+15%). More importantly,
it improves from 0.45 to 0.76 when considering daily observations (+68%). The daily offset is reduced by 60% from
OMI to TROPOMI (3.1 to $1.9 \times 10^{15}$ molec.cm$^{-2}$). In low-emission conditions, the MADs of the differences provide an
upper limit of the precision of the satellite measurements. If we consider HCHO levels below $8 \times 10^{15}$ molec.cm$^{-2}$
(medium level, but the low range is not represented here), the precision of the daily TROPOMI HCHO observations
is estimated to be $3 \times 10^{15}$ molec.cm$^{-2}$, which represents an improvement of more than a factor 2 compared to OMI. The
precision of monthly TROPOMI observations reaches $1.4 \times 10^{15}$ molec.cm$^{-2}$, which is close to the Copernicus user
requirements.



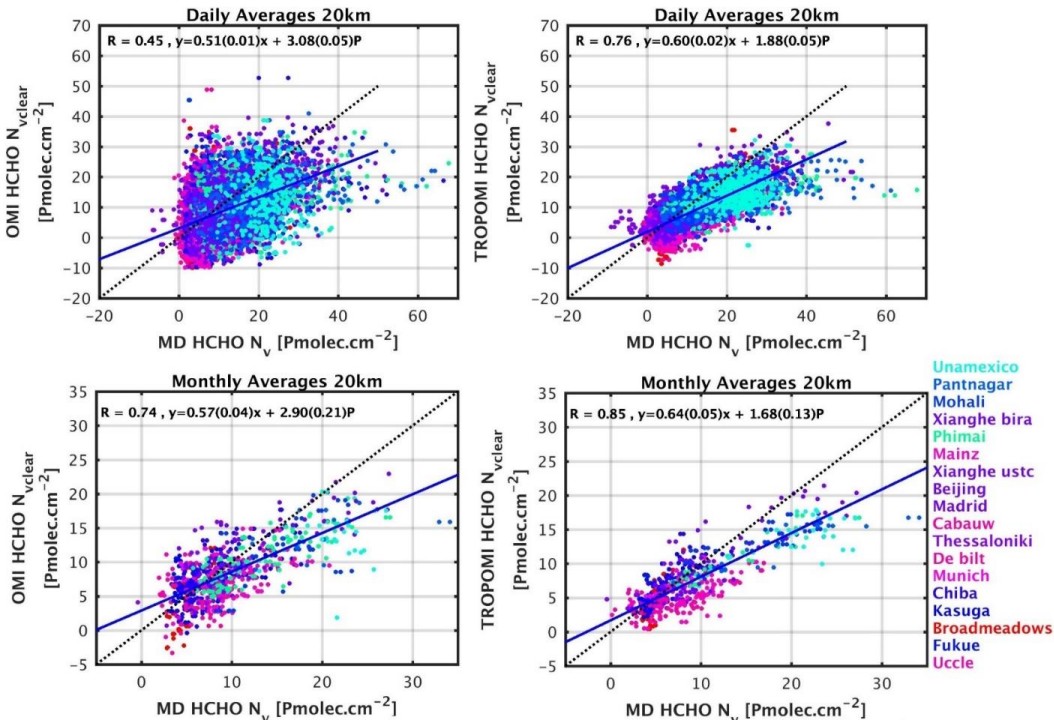

**556**

**Figure 15: Scatter plots of OMI (left) and TROPOMI (right) versus MAX-DOAS data for the daily (top) and monthly**
**(bottom) medians of collocated data. The correlation, slope and intercept of a linear regression using the robust Teil-Shein**
**estimator is given as inset and plotted as a blue line. The black dotted line is the 1:1 line. The color indicates the latitude of**
**the station. . [Pmolec.cm⁻²=10¹⁵ molec.cm⁻²].**
**Table 3: Summary of validation results for OMI and TROPOMI when considering all collocated pairs (daily or monthly**
**means) together. Values for HCHO columns lower or larger than 8x10¹⁵ molec.cm⁻² are given in brackets.**

|  | **OMI (<, >8x10¹⁵ molec.cm⁻²)** | **TROPOMI (<, >8x10¹⁵ molec.cm⁻²)** |
|---|---|---|
| **Daily** |  |  |
| MAD [10¹⁵ molec.cm⁻²] | 7.3 (6.7, 7.9) | 3.8 (3, 4) |
| Bias+-ErrB [%] | -18±7.5 (-7+-12,-21±6.9) | -11±3.6 (-10+-4.6, -25±2.8) |
| Offset [10¹⁵ molec.cm⁻²] | 3.1 | 1.9 |
| Slope | 0.51 | 0.6 |
| Correlation | 0.45 | 0.76 |
| **Monthly** |  |  |
| MAD [10¹⁵ molec.cm⁻²] | 2.6 (2.5, 3.2) | 2.3 (1.4, 2.7) |
| Bias+-ErrB [%] | -9±13 (9±16.6, -24±12) | -12±8.6 (-5±10, -25±5.7) |
| Offset [10¹⁵ molec.cm⁻²] | 2.9 | 1.7 |
| Slope | 0.57 | 0.64 |
| Correlation | 0.74 | 0.85 |

**5.2    Sensitivity tests**
We performed a few sensitivity tests, in order to evaluate the robustness of the validation results. First, we have used
different radii around the stations (from 10 to 100km), in order to detect possible spatial resolution effects. Results are
presented in Figure 16, for the TROPOMI case. At most stations, the bias shows marginally small dependency on the
radius. Again, this points to the large natural dispersion of the HCHO columns. We find an important exception at the



UNAM station in Mexico, where the bias clearly increases with the radius (-30% at 10km, -50% at 100km). At this
location, the correlation and MADs are also improved at 10km (not shown). In Beijing and Broadmeadows, we do
observe an increase of the bias at 100km resolution, but the values at 10 and 20km are mostly equivalent. We
performed the same test with OMI, and found consistent results, except that the lower sampling does not allow using
a 10km-radius area.

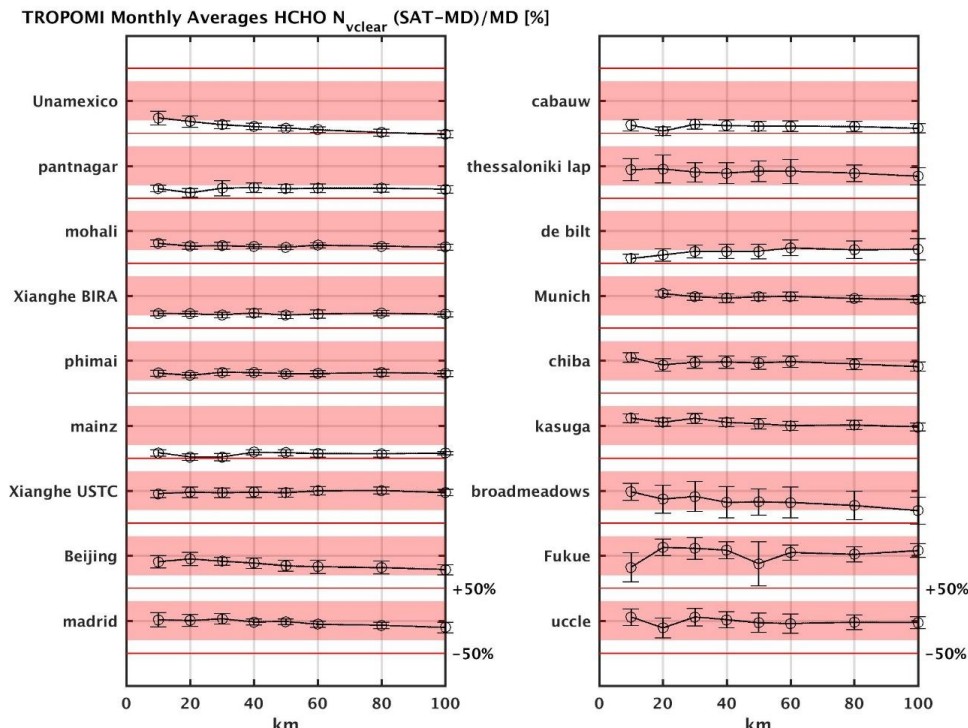

**Figure 16: Median monthly bias as a function of the radius taken around the validation sites. Pink areas indicate 40%
bias.**

We also evaluated the impact of clouds using two further tests: (1) compare the daily TROPOMI validation results for
$N_v$ and $N_{v\_clear}$, (2) use a much stricter cloud filter on cloud radiance fractions (CRF) of 20% instead of 60%
(equivalent to an effective cloud fraction of 10% instead of 40%). With this strict cloud filter, there is no difference
between $N_v$ and $N_{v\_clear}$. Results are summarized in Table 4. These tests indicate that the TROPOMI HCHO
validation results do not change significantly when a cloud correction is applied, although the $N_{v\_clear}$ results are
slightly better. Using a more stringent cloud filter reduces the number of observations. The bias for the lowest columns
becomes positive (from -10 to +3%), and the offset is increased (from 1.9 to $2.6 \times 10^{15}$ molec.cm$^{-2}$), while the negative
bias for the largest columns remains equivalent. These numbers will have to be re-evaluated using only the version 2
of the TROPOMI level 2 products available since July 2020, when enough data will be available. However, we note
that this limited impact of the cloud correction on the HCHO columns appears to be consistent with previous satellite





datasets, independently of the cloud product, as already observed with GOME-2 and OMI, using version 1 of the O2–
O2 cloud product (De Smedt et al., 2015).
**Table 4: Summary of daily validation results for TROPOMI when considering all collocated pairs when using $N_{v\_clear}$**
**(first column), (1) when using $N_v$ (second column) or (2) when using a strict cloud filter (third column).**

| | TROPOMI $N_{v\_clear}$ (<, >8x10$^{15}$ molec.cm$^{-2}$) | TROPOMI $N_v$ (<, >8x10$^{15}$ molec.cm$^{-2}$) | TROPOMI $N_{v\_clear}$ CRF<20% (<, >8x10$^{15}$ molec.cm$^{-2}$) |
|---|---|---|---|
| **Daily** | | | |
| MAD [10$^{15}$ molec.cm$^{-2}$] | 3.8 (3, 4) | 3.9 (3, 4.4) | 3.3 (2.6, 3.9) |
| Bias+-ErrB [%] | -11±3.6 (-10+-4.6, -25±2.8) | -14±-3.9 (-12±4.4,-29±2.9) | -3±4.6 (3±6.1, -27±3.8) |
| Offset [10$^{15}$ molec.cm$^{-2}$] | 1.9 | 1.8 | 2.6 |
| Slope | 0.6 | 0.56 | 0.57 |
| Correlation | 0.76 | 0.74 | 0.75 |

**5.3    Effect of vertical smoothing**
Three MAX-DOAS stations (Uccle, Xianghe BIRA-IASB, and UNAM) provide retrieved and a priori vertical profiles
together with corresponding averaging kernels (GEOMS format). This allows taking into account the different vertical
sensitivity of MAX-DOAS and TROPOMI measurements when making comparisons. We follow the methodology
from Rodgers and Connor (2003) described in detail in Vigouroux et al. (2020). It consists of two steps: first taking
into account the different a priori profiles used to retrieve these two data sets (Eq. 2 of Vigouroux et al., 2020), then
smoothing the ground-based profiles using TROPOMI averaging kernels (Eq. 3 of Vigouroux et al., 2020).
We give in Table 5 the MAD and biases obtained before and after application of the methodology, for the daily mean
comparisons. Note that the numbers at each site are slightly different than the ones obtained in sect. 5.1 (Figs. 5.3 and
5.5) because the collocated pairs are constructed slightly differently: each collocated pixel of the satellite must be
compared to MAX-DOAS before the daily average because the TROPOMI averaging kernel differs for each pixel.
We see in Table 5 that at the cleanest site (Uccle) the effect of the smoothing is small, while at the more polluted sites
Xianghe and UNAM, the biases are strongly reduced by about 20%. This result is in agreement with previous MAX-
DOAS validation studies (De Smedt et al., 2015; Wang et al., 2019b), but also with aircraft and regional model
comparisons (Zhu et al., 2020; Su et al., 2020). The effect of the smoothing is also clearly seen in Figure 17 where the
scatter plots of daily comparisons between TROPOMI and MAX-DOAS are shown before and after vertical
smoothing. The strong effect of the smoothing is usually not observed with FTIR comparisons because TROPOMI
and FTIR measurements have similar vertical sensitivity, which rapidly drops in the atmospheric layers lower than
3km (Vigouroux et al., 2020), while the MAX-DOAS shows an opposite sensitivity that is maximum at the surface
and generally becomes negligible above 3km (Vigouroux et al., 2008; De Smedt et al., 2015; Wang et al., 2019a).
This highlights the importance of taking into account the different a priori profiles and averaging kernels when
comparing techniques having different vertical sensitivity.
**Table 5: Effect of a priori substitution and vertical smoothing on the daily comparisons of TROPOMI and MAX-DOAS**
**data.**

| Daily | Direct comparisons | | Rodgers and Connor (2003) applied (a priori substitution and smoothing) | |
|---|---|---|---|---|
| | MAD [10$^{15}$ molec.cm$^{-2}$] | BIAS ± Err_B [%] | MAD [10$^{15}$ molec.cm$^{-2}$] | BIAS ± Err_B [%] |



| | | | | |
|---|---|---|---|---|
| **Uccle** | 2.4 | -9.4 ± 5.8 | 2.4 | -10.6 ± 5.5 |
| **Xianghe, BIRA** | 3.9 | -32.2 ± 2.5 | 2.7 | -9.1 ± 3.0 |
| **UNAM** | 6.1 | -34.3 ± 3.2 | 5.8 | -5.8 ± 5.7 |
| | **Scatter plot 3 sites** | | **Scatter plot 3 sites** | |
| Offset [$10^{15}$ molec.cm$^{-2}$] | 1.44 | | 0.29 | |
| Slope | 0.60 | | 0.88 | |
| Correlation | 0.84 | | 0.85 | |

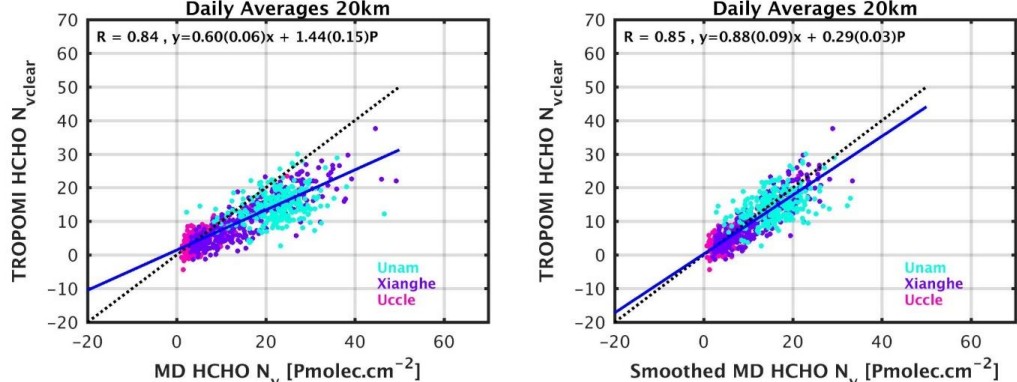

**Figure 17: Scatter plots of TROPOMI versus MAX-DOAS data for the daily means of collocated data before (left) and after (right) vertical smoothing of the MAX-DOAS profile in Uccle, Xianghe and UNAM/Mexico. The correlation, slope and intercept of a linear regression using the robust Teil-Shein estimator is given inset and plotted as a blue line. The black dotted line is the 1:1 line. [Pmolec.cm$^{-2}$=$10^{15}$ molec.cm$^{-2}$].**

## 6    Detection of weak HCHO columns over shipping lanes

As shown above, TROPOMI HCHO observations feature an unprecedented level of precision allowing for an improved detection of small columns at short time scales. Here, we present a case study to illustrate the ability of TROPOMI to detect small HCHO signals related to shipping emissions. When inspecting TROPOMI maps averaged over several months, weak lines of HCHO columns become visible over the background, especially in the Indian Ocean (see e.g. Figure 5). This becomes even clearer when saturating the continental HCHO columns by setting a lower maximum scale, as in Figure 18, which shows HCHO columns seasonally averaged over the months December, January and February between 2018 and 2021.

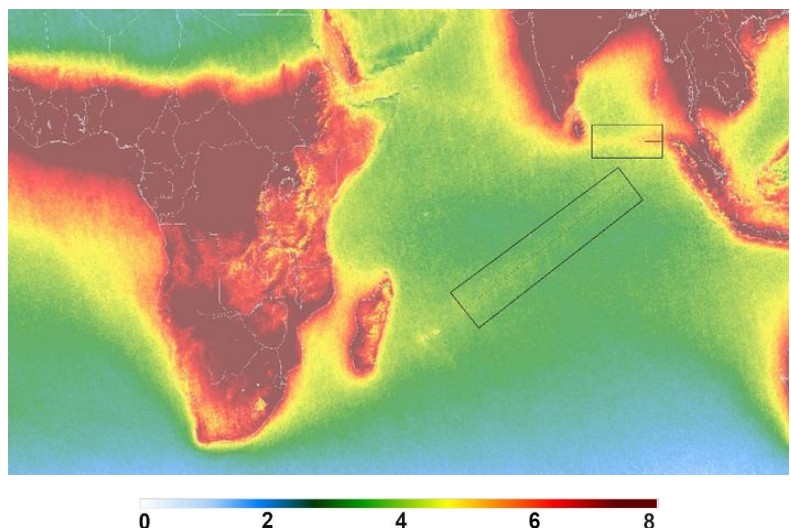

**Figure 18: Seasonal DJF map of TROPOMI HCHO tropospheric columns between Dec. 2018 and Feb.2021, on a spatial grid of 0.05° in latitude and longitude. Observations are only filtered using the provided qa_values >0.5. (max.scale: $8 \times 10^{15}$ molec.cm$^{-2}$).**

The detection of shipping emissions with satellite observations has often been reported for $NO_2$ (see for example Beirle et al., 2004; Richter et al., 2004; 2011; Boersma et al., 2015; Georgoulias et al., 2020), and more recently also for $SO_2$ based on OMI measurements (Theys et al., 2015). In the case of HCHO, however, only one study pointed to the identification of a shipping lane signal detected in a 7-year average of ERS-2 GOME data in the ship track corridor from Sri Lanka to Singapore (Marbach et al., 2009).

Here, we study two lines (1) from Sri Lanka to Singapore and (2) from Madagascar to Singapore. We perform an analysis and several sensitivity tests in order to gain confidence and information on the enhanced HCHO. As illustrated in the first panel of Figure 19 (line 1) and Figure 20 (line 2), in each box, we average the HCHO columns along the ship track to obtain a spatial cross section, and we bin the data as a function of the distance from the line (distances are expressed in degrees per 0.5° bin). The background level is not constant, for example due to continental outflow in the Bay of Bengal, and needs to be removed. To do so, we fit a straight line through the column values at the edges of the box and subtract this line from the signal. This allows to isolate a differential column and to evaluate its absolute and relative magnitude compared to the background (respectively shown in the second and third panels of Figure 19 and Figure 20. For comparison, we perform the same analysis using TROPOMI $NO_2$ tropospheric columns from the operational product (NO2 ATBD, Van Geffen et al., 2020). Although only about half as wide, the localisation of the $NO_2$ peak is found to be well aligned with the HCHO signal. Along the line from Sri Lanka to Singapore, we find a similar column enhancement and plume width as in Marbach et al. (2009).

In order to exclude a possible indirect AMF effect caused by the TM5 a priori profiles, the same analysis is done based on background-corrected slant columns (bc-SCD). We also restrict the analysis to clear sky observations, by using a strict cloud filtering of CRF<20%. Furthermore, we use the wind vector information provided in the TROPOMI L2 product from version 2 onwards (from August 2020), to select only clear-sky observations with low wind conditions



(qa>0.5, CRF<20%, W<5m/s). Finally, we add to the analysis a climatology of HCHO observations based on OMI
measurements (2005-2009).

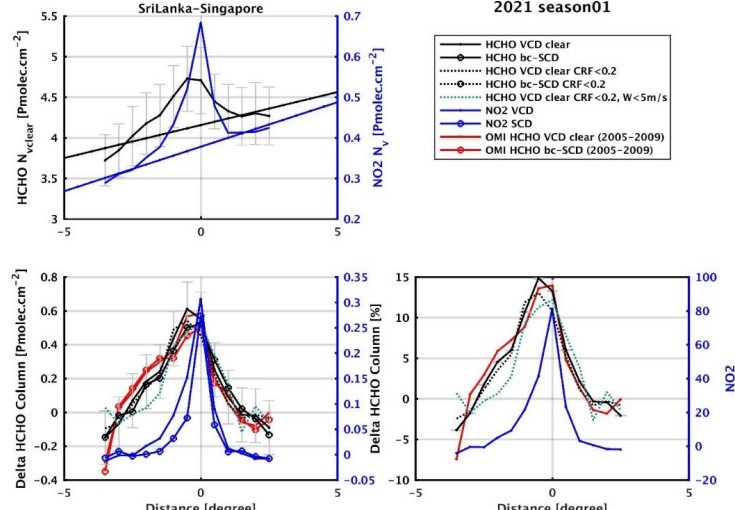


**Figure 19: Box average for the first selected line between Sri Lanka and Singapore between Dec. 2020 and Feb. 2021. The**
**x-axis represents the distance (south-north) in degrees from the shipping lane. The first panel shows the HCHO (in black)**
**and NO₂ (in blue) tropospheric columns, binned per distance from the line center. The fitted lines are used to remove the**
**background contribution. The two bottom panels present the absolute (left) and relative (right) column deviations from**
**the background line. The analysis is performed on the slant and the vertical columns (circles/lines), using a stricter cloud**
**filtering (CRF<20%, black dotted line), an additional filter on the wind velocity (W<5m/s, green dotted line), and finally on**
**OMI observations averaged between 2005 and 2009 (red). [Pmolec.cm⁻² = 1x10¹⁵ molec.cm⁻²].**

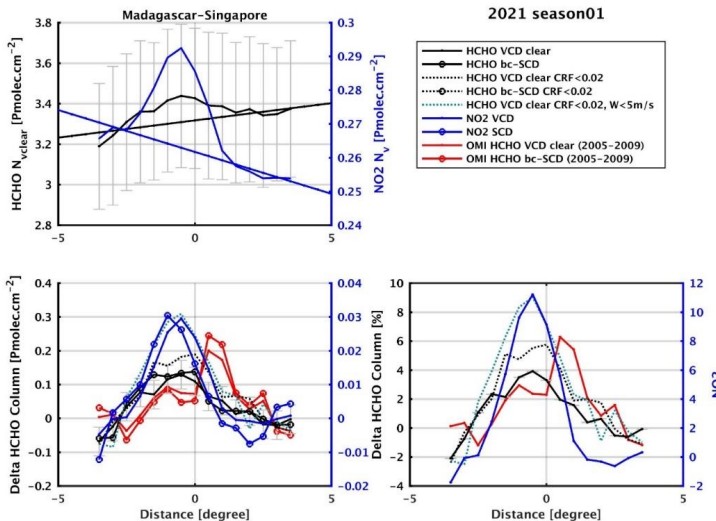


**Figure 20: Same as Figure 19 for the second selected line between Madagascar and Singapore.**





Using this approach, we analysed HCHO datasets for each season between MAM 2018 and DJF 2021. The absolute
and relative magnitude of the largest detected signal is plotted as a function of the season in Figure 21 and Figure 22.
Along the two lines, the signal is detected in the slant columns of HCHO and NO2 as well. This excludes the possibility
of an artefact coming from the TM5 a priori profiles. The signal remains detectable in clear-sky observations, and is
even increased along the second line. We observe a similar effect of the wind speed filtering (last two seasons).
Selecting only low-wind conditions clearly enhances the signal along line 2, and during SON along line 1. The
magnitude of the detected HCHO signal is larger along line 1 (from 0.2 to $0.7 \times 10^{15}$ molec.cm$^{-2}$, 15%) compared to
line 2 (from 0.1 to $0.3 \times 10^{15}$ molec.cm$^{-2}$, 8%). We find that the absolute magnitude of the HCHO signal is larger than
the $NO_2$ signal by a factor 3 to 10, but the relative increase of the $NO_2$ columns is significantly larger: 60% along line
1 and 15% along line 2. Both lines show a clear seasonality, particularly in the HCHO columns, with a maximum
during the DJF seasons seen in the OMI climatology and in the TROPOMI 3-months averages. The HCHO signal
presents a clear drop in JJA along line 1. This is related to the wind direction and strength, which bring the line signal
closer to the HCHO continental outflow, making its detection more difficult. The OMI data need to be averaged over
several years in order to detect a significant signal. While the first line is well detected in the 5-years OMI climatology,
the second line presents a smaller magnitude, a larger variability, and cannot be detected in the most recent years of
OMI measurements.

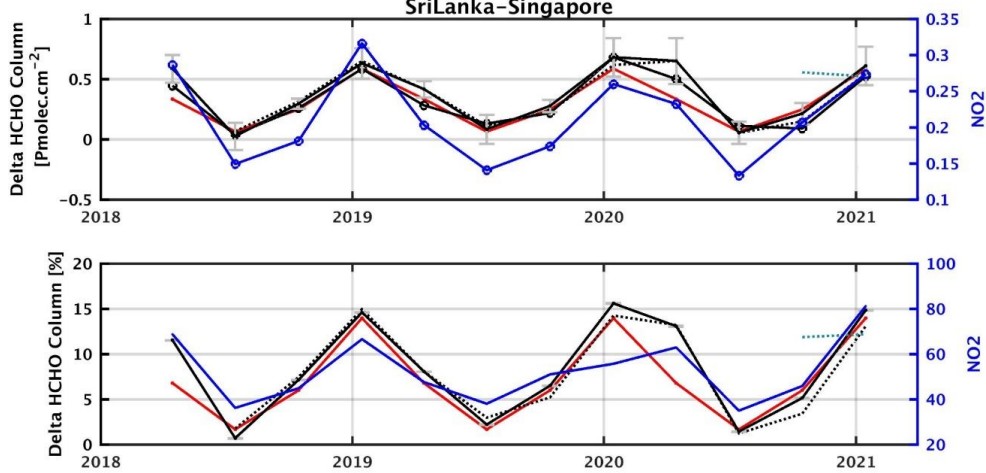


**Figure 21: Seasonal variation of the absolute (top panel) and relative (center panel) column deviations of the TROPOMI**
**HCHO (black), OMI 2005-2009 climatology HCHO (red) and TROPOMI NO₂ (blue) tropospheric columns along the Sri**
**Lanka – Singapore line. For each season, the maximum deviation compared to the background is provided. The results of**
**the analysis are given for the slant and the vertical columns (circles/lines), using a stricter cloud filtering (CRF<20%, black**
**dotted line), an additional filter on the wind velocity (W<5m/s, green dotted line). [Pmolec.cm⁻² = 1x10¹⁵ molec.cm⁻²].**

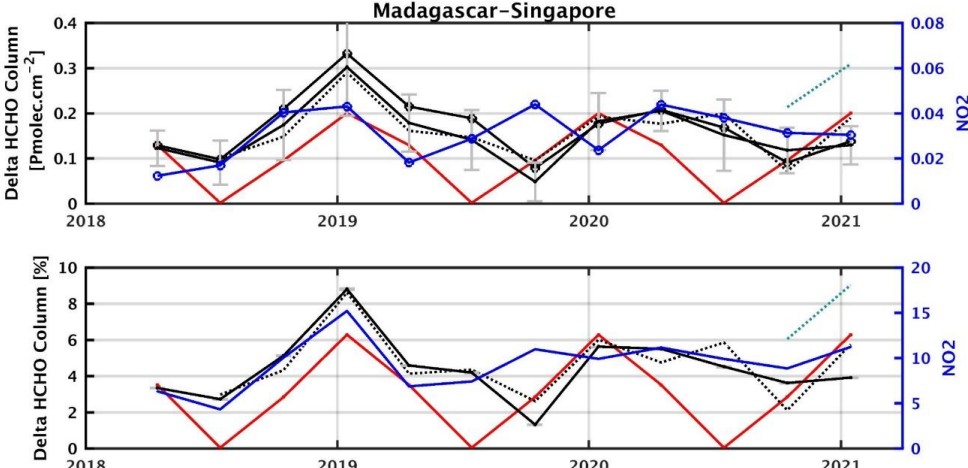

**Figure 22: Same as Figure 21 along the Madagascar – Singapore line.**

Using TROPOMI HCHO observations averaged over 3 months, it is therefore possible to detect a signal as small as

$0.1 \times 10^{15}$ molec.cm$^{-2}$ (with a median deviation of $0.03 \times 10^{15}$ molec.cm$^{-2}$), after removal of the background contribution.

Note that along the first line a similar analysis can also be performed on a monthly basis. While we show several

evidences that the signal is related to shipping emissions, its source is not studied here. As discussed in Marbach et al.

(2009) it could be due to secondary HCHO production via the atmospheric oxidation of NMVOCs emitted from ship

engines but also to enhanced $CH_4$ oxidation by elevated levels of OH radicals within the ship plumes. Model analysis

suggest that the second hypothesis is the main factor responsible for the elevated HCHO levels (Song et al.; 2010).

Other HCHO lines can be detected as well in the Tropics, although weaker in magnitude or closer to the continental

outflow (in the South-West of Africa or in the West of India). More advanced techniques to separate the signal from

the background and to account for wind dispersion effects could help in detecting more shipping lanes but also weak

continental emissions (Beirle et al., 2004).

## 7    Conclusions

Owing to its high spatial resolution resulting in many measurement points, coupled with an improved signal to noise

ratio at single pixel level, TROPOMI allows to monitor HCHO tropospheric columns from space with an

unprecedented definition. The global and regional maps show a clear reduction of the noise compared to previous

sensors, allowing for the detection of weaker HCHO signals, and the monitoring of HCHO variations on a much

shorter time scale.

We have evaluated the TROPOMI HCHO operational product against the QA4ECV OMI HCHO dataset, and against

a network of 18 ground-based MAX-DOAS instruments. The gain in precision at different spatial and temporal scales

was estimated by (1) comparing the median deviation of the averaged columns, and (2) validating the data using

MAX-DOAS column network measurements. Both methods include additional noise components from temporal



variation, spatial variation and ground-based column precision. Results are summarized in Figure 23 where precision
estimates are provided for observations over regions with enhanced continental emission and for background
conditions, as a function of the time resolution (daily or monthly averages) and of the spatial resolution (from 20km
to regional scale). At 20 and 100km resolution, both the median deviation approach and the validation results lead to
very consistent estimates of the precision. The theoretical noise is also represented in the figure; it decreases as the
squared root of the number of observations included in the averages. In remote conditions, the median deviation of
the averaged columns follows closely the theoretical noise until reaching a threshold. If we consider a large region in
the reference sector, all estimates converge towards a limit of about $0.2 \times 10^{15}$ molec.cm$^{-2}$ molec.cm$^{-2}$ (day) to $0.1 \times 10^{15}$
molec.cm$^{-2}$ (month) both for OMI and TROPOMI. Over continental emission sources, the reduction of the noise is
counterbalanced by the HCHO natural variability and by other source of pseudo-noise which depend on the spatial
and temporal scales of the observations. The largest improvement brought by TROPOMI is found for daily
observations at 20km resolution, for which a gain in precision by a factor of 3 is obtained compared to OMI. The
product and COPERNICUS user requirements for precision are also represented in the figure. Both are reached with
TROPOMI using daily averaged data at the resolution of 20km if we consider the dispersion in remote regions.
However, over continental emissions, local variability effects added up to the estimated precision that reaches a
threshold of about $2 \times 10^{15}$ molec.cm$^{-2}$.

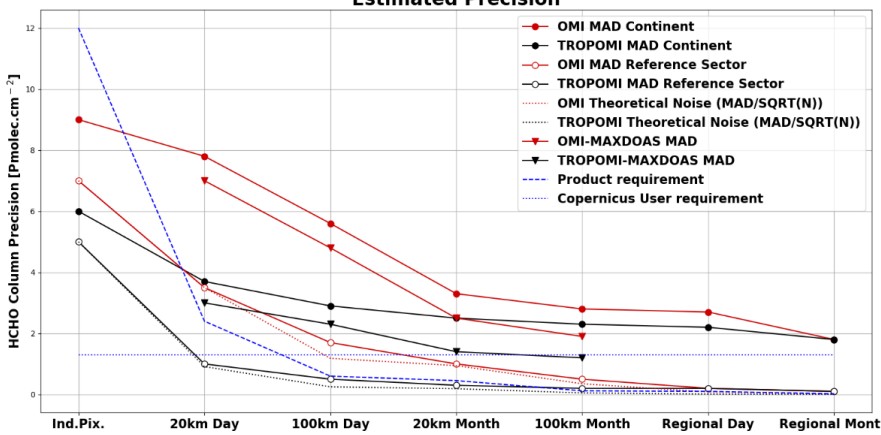

**Figure 23: Estimated precision of OMI (in red) and TROPOMI (in black) HCHO columns at different spatial and temporal**
**scales (20km, 100km, regions, day/month). The median deviation of the satellite HCHO columns are provided for**
**continental emissions (plain circles) and in the remote reference sector (white circles). Validation estimates are plotted at**
**20km and 100km (MAD of differences between satellite and MAX-DOAS columns, triangles). The theoretical noise (dotted**
**lines) corresponds to single measurement precision divided by the square root of observations. The dashed blue line is the**
**TROPOMI product requirement, based on a single measurement precision of $12 \times 10^{15}$ molec.cm$^{-2}$. The horizontal blue line**
**at $1.3 \times 10^{15}$ molec.cm$^{-2}$ represents the COPERNICUS user requirement. [Pmolec.cm$^{-2}$ = $1 \times 10^{15}$ molec.cm$^{-2}$].**
For the HCHO absolute values, we show that OMI and TROPOMI observations agree very well for moderate to large
HCHO levels (columns larger than $5 \times 10^{15}$ molec.cm$^{-2}$) for which the bias between both datasets is smaller than 10%.
For lower columns however, OMI observations present a remaining bias of about +20% compared to TROPOMI. This
good agreement is obtained by considering vertical columns calculated with air mass factors not corrected for cloud





effects (clear VCD). This allows to avoid biases related to differences in the cloud products. For all applications that
require combining the OMI and TROPOMI observations for low to moderate cloud fractions, we therefore advise to
use clear VCDs. Validation results confirm the good agreement between the OMI and TROPOMI datasets and a
similar underestimation of both products in the highest range of the HCHO levels (-25% in average for columns larger
than $8\times10^{15}$ molec.cm$^{-2}$). For medium columns, OMI presents a slight overestimation compared to MAX-DOAS data,
which is not observed for TROPOMI. Sensitivity tests show that validation results obtained with the TROPOMI
HCHO columns are weakly dependent on the cloud correction. They also depend weakly on the radius considered
around the station, with a few exceptions such as Mexico city or coastal stations. On the contrary, the vertical
smoothing (tested at three stations) has a strong effect on the comparison with MAX-DOAS. After taking into account
the different a priori profiles and averaging kernels, the bias for large HCHO columns is strongly reduced by about

748     20%.

Comparing OMI and TROPOMI monthly averaged HCHO columns, we do not observe significant differences related
to the spatial resolution, except in regions surrounded by natural boundaries where the benefit of the finer spatial
resolution of TROPOMI is clearly apparent. The weak sensitivity to the spatial resolution of HCHO measurements
can be understood when considering that HCHO is a secondary product from the degradation of NMOVCs with
various lifetimes, which results in a general spread of the HCHO spatial distributions. The large number of TROPOMI
observations allows to perform validation at a resolution as small as 10km on a daily basis with a sufficient precision,
which is not possible with OMI. It is clear that TROPOMI brings a significant improvement in the temporal resolution
of the observations. At most of the validation sites, TROPOMI allows for daily validation results as robust as those
obtained with OMI on a monthly basis.
The number of ground-based stations providing MAX-DOAS HCHO observations is constantly growing, providing
a large range of observation conditions, and for some of them, over several years allowing the comparisons of the
performances of several satellite datasets. Note however that the lower range of HCHO levels is under-represented,
as well as some of the largest emission regions such as South America or Africa. Following the validation study of
Vigouroux et al. (2020) based on a FTIR network of instruments, this study illustrates again the added value of using
a large network of instruments to draw more robust conclusions. FTIR and MAX-DOAS networks are complementary
to each other and could be combined to cover as many conditions as possible. Similarly to what was achieved for the
FTIR network, the MAX-DOAS HCHO datasets would benefit from further homogenisation efforts.
Finally, to illustrate the benefit of TROPOMI for the detection of small HCHO signals, we present a case study
addressing the detection of shipping lanes in the Indian Ocean. Using simultaneous observations of tropospheric $NO_2$
and meteorological wind field data, we present strong evidences for an HCHO production in regions affected by
shipping emissions. Owing to the sensitivity of TROPOMI, such small signals can now be observed from space on a
seasonal basis.
**Code and data availability**



The S5p HCHO data are available at https://scihub.copernicus.eu. The access and use of any Copernicus Sentinel data available
through the Copernicus Sentinel Data Hub is governed by the Legal Notice on the use of Copernicus Sentinel Data and Service
Information and is given here: https://sentinels.copernicus.eu/documents/247904/690755/Sentinel_Data_Legal_Notice.
The QA4ECV OMI HCHO product is available at https://doi.org/10.18758/71021031 (De Smedt et al., 2017). The MAX-DOAS
datasets can be requested from the individual PIs of each station.

## Author contributions

IDS coordinated the paper and carried out the analysis. GP and CV are PIs of the NIDFORVAL S5PVT project, SC ensures the
MPC routine validation. IDS, PH, YH, CLe, DL, FR, NT, JV, MVR developed the TROPOMI HCHO product. FB, IDS, YH, AR,
MVR, TW developed the QA4ECV OMI HCHO product. AB, NB, KLC, SD, FH, HI, VK, CLi, AP, CRC, RGR, MVR, TW are
PIs for the QA4ECV MAX-DOAS measurements. BL, SC, GP, CV performed MAX-DOAS data collection and format
harmonization and carried out the validation analysis. SC, KUE and JCL are responsible of the MPC routine validation. MVR is
the coordinator of this research. All co-authors revised and commented on the paper.

## Acknowledgements

This work contains modified Copernicus Sentinel-5 Precursor satellite data (2018-2020) post-processed by BIRA-IASB. Part of
the reported work was carried out in the framework of the Copernicus Sentinel-5 Precursor Mission Performance Centre (S5p
MPC), contracted by the European Space Agency (ESA/ESRIN, Contract No. 4000117151/16/I-LG) and supported by the Belgian
Federal Science Policy Office (BELSPO), the Royal Belgian Institute for Space Aeronomy (BIRA-IASB) and the German
Aerospace Centre (DLR). BIRA-IASB acknowledges national funding from BELSPO and ESA through the ProDEx projects
TRACE-S5P (TRACE-S5P project) and TROVA. Part of this work was carried out also in the framework of the S5p Validation
Team (S5PVT) AO projects NIDFORVAL (ID #28607, PI G. Pinardi, C. Vigouroux, BIRA-IASB). Multi-sensor HCHO
developments have been funded by the EU FP7 QA4ECV project (grant no. 607405), in close cooperation with KNMI, University
of Bremen, MPIC-Mainz and WUR. Work by H. Irie was supported by the Environment Research and Technology Development
Fund (JPMEERF20192001 and JPMEERF20215005) of the Environmental Restoration and Conservation Agency of Japan, JSPS
KAKENHI (grant numbers JP19H04235 and JP20H04320), and the JAXA 2nd research announcement on the Earth Observations
(grant number 19RT000351). We acknowledge Mark Wenig from LMU for supporting the MAX-DOAS operations in Munich,
Caroline Fayt and Christian Herman from BIRA-IASB for the Uccle and Xianghe instruments, as well as Pucaï Wang from
IAP/CAS for maintaining the BIRA-IASB instrument in Xianghe. We thank Alejandro Bezanilla from CCA-UNAM, Manish Naja
from ARIES for the MAX-DOAS instrument operation in Pantnager and Thanawat Jarupongsakul from Chulalongkorn University
for the Phimai station. We acknowledge IISER Mohali Atmospheric Chemistry Facility for supporting the MAX-DOAS operations
in Mohali.

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
