# Peer review of "Comparative assessment of TROPOMI and OMI formaldehyde"

_Atmospheric Chemistry and Physics, 2021_

## Author Comment (AC1)

**We would like to thank a lot the reviewers for their very positive comments and good suggestions. Thank you for taking the time to read it carefully and apologies for the length of the paper. We have considered all the comments. See our detailed answers hereafter.**

**Answer to Anonymous Referee #1**

The vertical sensitivity of OMI and TROPOMI on individual scenes are unfortunately not discussed. It would have been very interesting to compare, e.g., the OMI and TROPOMI averaging kernels in typical observational conditions in order to know whether TROPOMI allows probing the tropospheric layers closer to the surface. Although the manuscript is already quite long, perhaps a discussion can be added in this regard.

The vertical sensitivity of OMI and TROPOMI is very similar. The observation angles and overpass times are the same. We do not expect any significant differences. Only the selection of the pixels could lead to small differences, since a number of nadir OMI observations have to be discarded because of the row anomaly. We added this information in the paper (line 609). We have added a figure in the supplement showing typical averaging kernels for OMI, TROPOMI and the MAX-DOAS instrument in Xianghe.

"We see in **Error! Reference source not found.** that at the cleanest site (Uccle) the effect of the smoothing is small, while at the more polluted sites Xianghe and UNAM, the biases are strongly reduced by about 20%. This result is in agreement with previous MAX-DOAS validation studies (De Smedt et al., 2015; Wang et al., 2019b), but also with aircraft and regional model comparisons (Zhu et al., 2020; Su et al., 2020). The effect of the smoothing is also clearly seen in **Error! Reference source not found.** where the scatter plots of daily comparisons between TROPOMI and MAX-DOAS are shown before and after vertical smoothing. The strong effect of the smoothing is usually not observed with FTIR comparisons because TROPOMI and FTIR measurements have similar vertical sensitivity, which rapidly drops in the atmospheric layers lower than 3km (Vigouroux et al., 2020), while the MAX-DOAS shows an opposite sensitivity that is maximum at the surface and generally becomes negligible above 3km (Vigouroux et al., 2008; De Smedt et al., 2015; Wang et al., 2019a). An illustration of typical averaging kernels for OMI, TROPOMI and the MAX_DOAS instrument in Xianghe is provided in Figure S6. As the observation angles and overpass times are very close for OMI and TROPOMI, their measurements come with a similar vertical sensitivity. This highlights the importance of taking into account the different a priori profiles and averaging kernels when comparing techniques having different vertical sensitivity."

[Figure]

**Figure S1: Typical column averaging kernels for TROPOMI, OMI and MAXDOAS instruments on 1 July 2020 over Xianghe.**

It is clear that cloud filtering and cloud correction are crucial for ensuring satisfactory HCHO retrievals from the satellite sensors. On a reference sector, for a typical day, what is the proportion of OMI and TROPOMI measurements that are discarded because of the cloud coverage or that are affected by residuals clouds? For example, how many measurements fall into, respectively, the CF<0.1, 0.1<CF<0.4 and CF>0.4 categories? I also assume that the MAX-DOAS observations are affected similarly by the presence of clouds. Therefore, are the MAX-DOAS measurements filtered out or corrected the same way? I see on Figs 15 and 17 that the OMI and TROPOMI clear-sky columns (Nv_clear) are compared with the MAX-DOAS measurements (called Nv). Are these latter also calculated using clear-sky AMF? If no, does it affect the comparison with the satellite data?

For a typical day in the reference sector, we find the following proportions of observations:

- for TROPOMI :
    - CF<0.1: 55%
    - 0.1<CF<0.4: 30%
    - CF>0.4: 15%
- For OMI:
    - CF<0.1: 65%
    - 0.1<CF<0.4: 25%

○ CF>0.4: 10%

These estimates depend on the cloud product used for each instruments, but they show that more than half of the observations are generally cloud free.

The selection on the MAX-DOAS observations is done based on the satellite measurements. Only the coincident observations within +-3h are kept in the comparisons. We therefore assume that the MAX-DOAS observations include only low cloud coverages. The great majority of the MAX-DOAS retrievals used in this study do not include a cloud correction. There is therefore no distinction between VCD and VCD clear for the ground-based observations.

Lines 241-245: Perhaps there is something I have misunderstood here, but it is relatively unclear to me why and how the OMI observations are daily gridded on a grid with a resolution as high as 0.05 x 0.05°. In the tropics, such a grid cell represents roughly a 5 x 5 km area, even smaller at higher latitudes, whereas the typical OMI pixel size is much larger than that. Hence, a single OMI pixel should overlap several grid cells. How do you attribute, on a daily basis, a value to such a small grid cell with observations having a footprint of 13 x 24 km in the best case? Without overlaps between the OMI observations taken the same day, I guess that oversampling techniques cannot be used. As well, what do you mean by *"we require the region to be filled with at least 50% of valid grid cells"* ? How can you have a minimum of 2 OMI observations per 0.05 x 0.05° grid cell on a daily basis?

For the 0.05 grids: the gridding routine (HARP: https://stcorp.github.io/harp/doc/html/index.html) attributes a weight to the satellite pixel, according to the fraction of the surface pixel that covers the grid cell, which is less than for OMI.

For the regions: Here we mean the large regions represented by the black boxes on the map. We require that half of the grid cells included in the boxes contain non-default values.

Lines 336-337: If I understood correctly, the random errors on the tropospheric vertical column is driven by the error on the slant column. However, I assume that the AMF calculation is also affected by uncertainties, which in turn would convolve with the SCDE and add to the total uncertainty on the vertical column. Is that accounted for?
Yes, the total uncertainty on the vertical columns include errors coming from the slant columns, the air mass factors and the background correction estimations (see De Smedt et al., 2015 or TROPOMI ATBD). However, the random error on a single observation is largely dominated by the slant column random error. Therefore, for the precision estimates in this paper, we focus on this part of the error component. AMF and background correction errors are largely systematic, and are considered in the bias estimates.

Lines 529-539: Despite the underestimation, the results of the comparison with the BIRA-IASB instrument at Xianghe are actually more in line with the comparisons shown by Vigouroux et al. (2020) and the overall underestimation of the larger columns by TROPOMI. Perhaps it is worth to be mentioned.

Yes, indeed. We added a sentence here, thank you. (Line 533: "However, this larger bias is in better agreement with the results found for equivalent stations in India and with FTIR validation results in Xianghe (Vigouroux et al., 2018).")

Figure 20: Can you explain why the second ship line is detected more to the North by OMI? Is it because the OMI 2005-2009 dataset used here is an annual mean, whereas the TROPOMI data are selected between Dec. 2020 and Feb. 2021?

We cannot give a firm explanation. It could be that the shipping lane has actually been displaced along the years. But the poorer detection using the OMI data of the more recent years does not allow to confirm this hypothesis. The misalignment between the OMI climatology and the TROPOMI signal could also be due to artefacts in the OMI

detection along this second, weaker, line. The coarser spatial resolution of OMI compared to TROPOMI could also play a role in the direction of this line.

Figure S1, right panel: I am a bit puzzled to see that big cities in Western and Northern

Europe (e.g., London, Paris, the Ruhr Basin, etc.) are not visible whereas cities located

more to the South around the Mediterranean Sea clearly are (e.g., Marseille, Barcelona,

etc.). Anthropogenic emissions of HCHO precursors should also be quite high in

Western and Northern Europe. Similarly, it looks like the annual HCHO columns are at

least as high over the Sahara Desert, far from any direct sources, as over, e.g., England and most of Western Europe. Is there a latitudinal dependence of the retrievals that would introduce a bias?

There is indeed a clear latitudinal dependency of the HCHO columns, which is mostly due to the temperature variations. Temperature and sunshine rate are the main drivers of the HCHO production, including the methane oxidation source. For a similar NMVOC emission, the HCHO production yield will increase with temperature.

Line 32: assesses
ok
Line 49: a factor of 3
ok
Lines 56-61 and further in the text: There are several species that are not defined
when they are first used. E.g., nitrogen oxides (NOx), carbon monoxide (CO), etc.
corrected
Line 63: Because of
ok
Line 91: Do you mean a daily average on a global 20 x 20 km resolution grid or in a 20
km-radius area around a specific point?
This is not specified in the scientific requirements. Only 20km spatial resolution is mentioned; so this is indeed subject to interpretations.
Line 97: requirements
ok
Lines 121-122: On board Aura
ok
Line 138: On board the S5P
ok
Line 273: 3-year averages
ok
Line 315: I think using "*3-year*" instead of "*long-term*" here is more appropriate.
ok
Figure 5: The color bar of the bottom panel could be better adapted; the differences are
currently difficult to distinguish.
ok
Line 423: I would avoid using "`HCHO emissions`", knowing that HCHO is mainly
secondarily formed in the atmosphere.
ok
Line 445: see Table 2
ok
Line 448: 20 km radius?
ok
Line 503: Mexico, respectively.
ok

Line 523: I suggest using "*relative*" instead of "*compared*"
ok
Table 3, 4th row: Must be "±" instead of "+" in the two columns
ok
Line 645: Figure 20).
ok
Line 673: a factor of 3
ok
Line 678: 5-year OMI climatology
ok
Line 695: suggests
ok
Line 711: emissions
ok
Line 717: Delete one of the "*molec. cm-2*"
ok
Line 719: sources
ok
Line 769: "*Owing to the sensitivity of TROPOMI*". I disagree. It is primarily a matter of spatial resolution and high spatial sampling, not of sensitivity. For example, it is said earlier that such shipping lanes could already be detected with GOME, combining many years of measurements.
We agree with you and changed the sentence. Note that the GOME spectral resolution was 0.2nm, significantly better than OMI or TROPOMI (0.5nm). This could play a role in the sensitivity as well.
Several references are not up-to-date (e.g., still refer to papers in discussion whereas the final papers have already been published).

OK, thanks!

**Answer to Anonymous Referee #2**

An (optional) general suggestion would be to change the title somewhat. I was expecting a purely validation paper, but the first half is almost entirely assessing the instruments and looking at relative OMI and TROPOMI performance. I wonder if the current title could cause the paper to be passed over by people who don't want to read another validation paper, but are still interested generally in TROPOMI details and TROPOMI vs OMI performance. Maybe something like: "Comparative assessment of TROPOMI and OMI formaldehyde observations and validation against MAX-DOAS network column measurements". There is a lot of useful information in here that stands alone without the validation analysis.

Thank you for the suggestion. We adopt this new title.

**Specific Comments**
Line 53: Add time period for previous sensors for context, for example: "previous sensors (X years)"
ok.
Line 103: Suggest change "high emission" to "high concentration" or similar (since emission maybe be from elsewhere, not emitted at site)
ok
Line 187: I'm a bit confused about the correction introduced here... maybe I just need a more detailed explanation. I gather it is applied to TROPOMI and OMI for a more direct comparison throughout this paper? Are there possible negative consequences of doing this, for instance in the interpretation of results by users, that need to be discussed here?

The correction is indeed applied to OMI and TROPOMI data. The implications are discussed in the paper, in section 4.2 for OMI/TROPOMI comparisons, and then in section 5.2 for validation.

For OMI/TROPOMI comparisons, there are actually only advantages to do so, except of course for very cloudy pixels, but they are anyway excluded by the QA filtering (CF<0.4). We advised the users of OMI-TROPOMI combined time series to apply this transformation. We further show that the TROPOMI validation results are not improved if a cloud correction is applied, since VCD are not significantly different from VCD clear for moderated cloud fractions (between 0.1 and 0.4). In contrast, the OMI VCD clear tend to show more differences compared to the OMI VCD.

**Technical Comments**
Line 243: Better to say "Throughout the paper" instead of "Along the paper"
ok
Line 252: Slightly confusing about how k is applied… Is the equation k*median(..) and text is supposed to be k=1.4826?
The factor k =1.4826 is applied on the median of the differences, yes.
Line 283: Change "to detect" to "the detection of"
ok
Line 445: missing statement after "see"

Corrected.